# NINJ1 mediates inflammatory cell death, PANoptosis, and lethality during infection conditions and heat stress

Joo-Hui Han [1,5,8], Rajendra Karki [1,6,8], R. K. Subbarao Malireddi [1], Raghvendra Mall [1,7], Roman Sarkar[1], Bhesh Raj Sharma[1], Jonathon Klein[2], Harmut Berns [2], Harshan Pisharath[3], Shondra M. Pruett-Miller [2], Sung-Jin Bae[4] & Thirumala-Devi Kanneganti [1] ✉

Innate immunity provides the first line of defense through multiple mechanisms, including pyrogen production and cell death. While elevated body temperature during infection is beneficial to clear pathogens, heat stress (HS) can lead to inflammation and pathology. Links between pathogen exposure, HS, cytokine release, and inflammation have been observed, but fundamental innate immune mechanisms driving pathology during pathogen exposure and HS remain unclear. Here, we use multiple genetic approaches to elucidate innate immune pathways in infection or LPS and HS models. Our results show that bacteria and LPS robustly increase inflammatory cell death during HS that is dependent on caspase-1, caspase-11, caspase-8, and RIPK3 through the PANoptosis pathway. Caspase-7 also contributes to PANoptosis in this context. Furthermore, NINJ1 is an important executioner of this cell death to release inflammatory molecules, independent of other pore-forming executioner proteins, gasdermin D, gasdermin E, and MLKL. In an in vivo HS model, mortality is reduced by deleting NINJ1 and fully rescued by deleting key PANoptosis molecules. Our findings suggest that therapeutic strategies blocking NINJ1 or its upstream regulators to prevent PANoptosis may reduce the release of inflammatory mediators and benefit patients.

The innate immune system acts as the critical first line of defense against infection and disease. Innate immunity uses pattern recognition receptors (PRRs) to detect pathogens, pathogen-associated molecular patterns (PAMPs), and damage-associated molecular patterns (DAMPs) and activate signaling cascades. One of the key outcomes of this signaling is the production of cytokines and danger signals[1], some of which act as endogenous pyrogens, such as IL-1, TNF,

IL-6, and HMGB1[2,3]. Pyrogens induce elevated body temperature (fever), which is a hallmark of infection and inflammatory disease[4]. In general, fever is associated with improved outcomes during infection[5-8]. However, fever is not universally beneficial, and uncontrolled fever is associated with worse outcomes in patients with sepsis or neurological injuries[9]. In addition, elevated body temperature and heat stress (HS) are associated with heat stroke, a life-threatening

[1]Department of Immunology, St. Jude Children's Research Hospital, Memphis, TN 38105, USA. [2]Center for Advanced Genome Engineering, St Jude Children's Research Hospital, Memphis, TN 38105, USA. [3]Animal Resources Center, St Jude Children's Research Hospital, Memphis, TN 38105, USA. [4]Department of Molecular Biology and Immunology, College of Medicine, Kosin University, Busan 49267, Republic of Korea. [5]Present address: College of Pharmacy and Research Institute of Pharmaceutical Sciences, Woosuk University, Wanju 55338, Republic of Korea. [6]Present address: Department of Biological Sciences, Seoul National University, Seoul 08826, Republic of Korea. [7]Present address: Biotechnology Research Center, Technology Innovation Institute, Abu Dhabi, P.O. Box 9639, United Arab Emirates. [8]These authors contributed equally: Joo-Hui Han, Rajendra Karki. ✉e-mail: Thirumala-Devi.Kanneganti@StJude.org

condition with a systemic inflammatory response leading to multi-organ dysfunction[10]. Elevation of core body temperature in lipopolysaccharide (LPS)-challenged rats substantially augments circulating levels of pyrogenic cytokines[11], suggesting that underlying illness or infection exacerbates HS responses on a cellular level. Furthermore, with prolonged hyperthermia, heat-related cytotoxic effects cause dysregulated inflammation[12,13]. While many links between HS, cytokine release, and inflammation have been observed, the fundamental molecular mechanisms of innate immune activation driving pathological responses to HS remain unclear.

In addition to inducing the production of cytokines and danger signals for inflammation, innate immune activation can also drive cell death[14]. Cell death causes inflammation through the activation of pore-forming cell death executioners, which form holes in the plasma membrane and allow the release of inflammatory mediators, including cytokines and DAMPs. Diverse cell death pathways are associated with the activation of different executioners, such as the gasdermin family of proteins and MLKL[15]. Cell death has also been observed in innate immune cells in response to HS[16]. However, the specific cell death pathways involved and the role of these pore-forming cell death executioners in this process are not well understood.

Therefore, in this study, we evaluate the innate immune response to HS, with a focus on the role of inflammatory cell death. HS induces inflammatory cell death, PANoptosis, that is exacerbated by the presence of pathogens or PAMPs. Leveraging CRISPR screening technology, our results show that HS and PAMP-induced PANoptosis occurs through the executioner NINJ1. In line with these data, in vivo protection from HS and PAMP-mediated mortality is provided by deletion of NINJ1 or combined deletion of other PANoptosis-associated molecules. Overall, our results suggest NINJ1 is a critical executioner in HS plus PAMP-induced PANoptosis to drive pathology and mortality, implicating molecules in the PANoptosis pathway as potential targets for therapeutic intervention.

## Results

### HS induces NLRP3 inflammasome activation during infection

Exogenous heat is known to induce cytotoxicity[16], but the innate immune pathways involved in this process are unclear. In addition, the current paradigm in the pathophysiology of heatstroke and fevers places heat as the primary trigger and driver of HS. However, pathological and clinical reports from patients with HS and studies from preclinical models suggest that heat alone does not fully explain the pathophysiology[17], and underlying illness or infection increases the susceptibility to sepsis during HS[18]. Therefore, to investigate the role of innate immunity and cell death in response to HS with or without infection, we incubated bone marrow-derived macrophages (BMDMs) at 43 °C to induce HS with or without pathogens or PAMPs, and then monitored cell death over time following removal of the stress (Fig. 1A). Being subjected to HS for 30 min induced a low level of cell death in BMDMs (Fig. 1B, C). Infection with *Escherichia coli* or *Citrobacter rodentium* increased the incidence of cell death during HS compared with infection without HS (Fig. 1D, E). To determine whether whole pathogens were required for this effect, we also tested cell death in response to LPS, which is one of a major PAMPs in the outer membrane of bacteria that can be released in the host during infection[19]. LPS from gut microbiota can translocate to the blood as an endotoxin during hyperthermia[9], and we observed that LPS treatment with HS in BMDMs increased the incidence of cell death (Fig. 1F, G). Similar findings were observed in human macrophages, human monocytes, THP-1 cells, RAW264.7 cells, and L929 cells (Supplementary Fig. 1), indicating that HS can induce cell death in multiple cell types of mouse and human origin during infection. Together, these results suggest that bacterial infection or LPS treatment potentiates the cell death induced by HS.

Innate immune sensors play critical roles in activating inflammatory signaling pathways and driving cell death. To identify the upstream innate immune sensors and regulators that mediate cell death in response to PAMP plus HS treatment, we first examined the cell surface PRRs and assessed the effect of multiple Toll-like receptors (TLRs) in promoting the cell death observed during HS. We primed BMDMs with Pam3CSK4 (TLR1/2 agonist), poly(I:C) (TLR3 agonist), LPS (TLR4 agonist), imiquimod (TLR7 agonist), or CpG (TLR9 agonist) to engage specific TLR signaling for 2 h before subjecting the cells to HS for 30 min. BMDMs primed with poly(I:C) or LPS, but not Pam3CSK4, imiquimod, or CpG, underwent robust cell death in response to HS (Supplementary Fig. 2A, B). Poly(I:C) and LPS signal through TLR3 and TLR4, respectively, which both mediate downstream signaling through the adaptor molecule TRIF, with TLR3 signaling exclusively through TRIF and TLR4 signaling through both TRIF and MyD88. To assess whether TRIF played a role in this cell death, we primed *Trif*[−/−], *Myd88*[−/−], or *Trif*[−/−]*Myd88*[−/−] BMDMs with LPS and subjected them to HS. *Trif*[−/−] and *Trif*[−/−]*Myd88*[−/−] BMDMs, but not *Myd88*[−/−] BMDMs, showed reduced cell death in response to LPS + HS (Supplementary Fig. 2C, D). In contrast, we found that the deletion of TRIF did not reduce the cell death caused by HS alone (Supplementary Fig. 2E, F), indicating that infection or specific PAMP stimulation in conjunction with the HS activates distinct cell death mechanisms compared to HS alone. Overall, these findings suggest TLR signaling through TRIF enhances cell death triggered by HS during infection.

We next investigated the involvement of the cytosolic sensors that are known to assemble inflammasomes by testing for caspase-1 activation and cell death, as inflammasome formation is known to occur downstream of TLR activation[20]. The caspase-1 cleavage induced in response to LPS plus HS treatment was abolished in *Nlrp3*[−/−] and *Pycard*[−/−] BMDMs, but not in *Aim2*[−/−], *Nlrp1b*[−/−], *Nlrc4*[−/−], *Mefv*[−/−], or *Casp11*[−/−] BMDMs (Supplementary Fig. 3A), indicating that NLRP3 senses homeostatic alterations induced by LPS plus HS treatment to activate the inflammasome. Despite the loss of caspase-1 cleavage (Supplementary Fig. 3A), cell death was not impaired in *Nlrp3*[−/−] or *Pycard*[−/−] BMDMs (Fig. 1H and Supplementary Fig. 3B), suggesting that other molecules are involved in regulating cell death in this context. We further screened other innate immune sensors that are known to regulate inflammasome activation and cell death in various contexts, including NLRC1, NLRC2, NLRC3, NLRC5, NLRP6, and NLRP12. None of these sensors were found to regulate cell death in response to LPS plus HS (Fig. 1I, and Supplementary Fig. 3C).

NLRP3 inflammasome activation and cell death can also be induced by the innate immune sensor ZBP1[21,22]. In addition, it was previously suggested that ZBP1 can drive cell death in response to HS[16]. However, we observed similar dynamics of cell death in wild type (WT), *Zbp1*[−/−], and *Zbp1*[ΔZα2] BMDMs, as well as in control siRNA-treated and *Zbp1* siRNA-treated BMDMs, in response to HS in the presence or absence of LPS (Fig. 1J–L and Supplementary Fig. 4A). To further confirm our findings, we used CRISPR to generate a new *Zbp1*[−/−] mouse line (referred to here as *Zbp1*[TDK]) (Supplementary Fig. 4B, C). *Zbp1*[TDK] BMDMs showed a similar rate of cell death with respect to WT BMDMs treated with HS in the presence or absence of LPS (Supplementary Fig. 4D–G). ZBP1 has also been reported to co-operate with Pyrin (*Mefv*) to mediate inflammatory cell death during *Francisella* and HSV1 infections[23]. However, we observed a similar extent of cell death in *Zbp1*[−/−] and *Zbp1*[−/−]*Mefv*[−/−] BMDMs treated with or without the NLRP3 inhibitor MCC950 with respect to WT BMDMs (Supplementary Fig. 4H), suggesting that ZBP1 did not co-operate with Pyrin or NLRP3 to drive the cell death. Altogether, our findings suggest that HS induces NLRP3 inflammasome activation during infection, but that NLRP3, ZBP1, and the other innate immune sensors tested are not required to induce the inflammatory cell death caused by HS and infection.

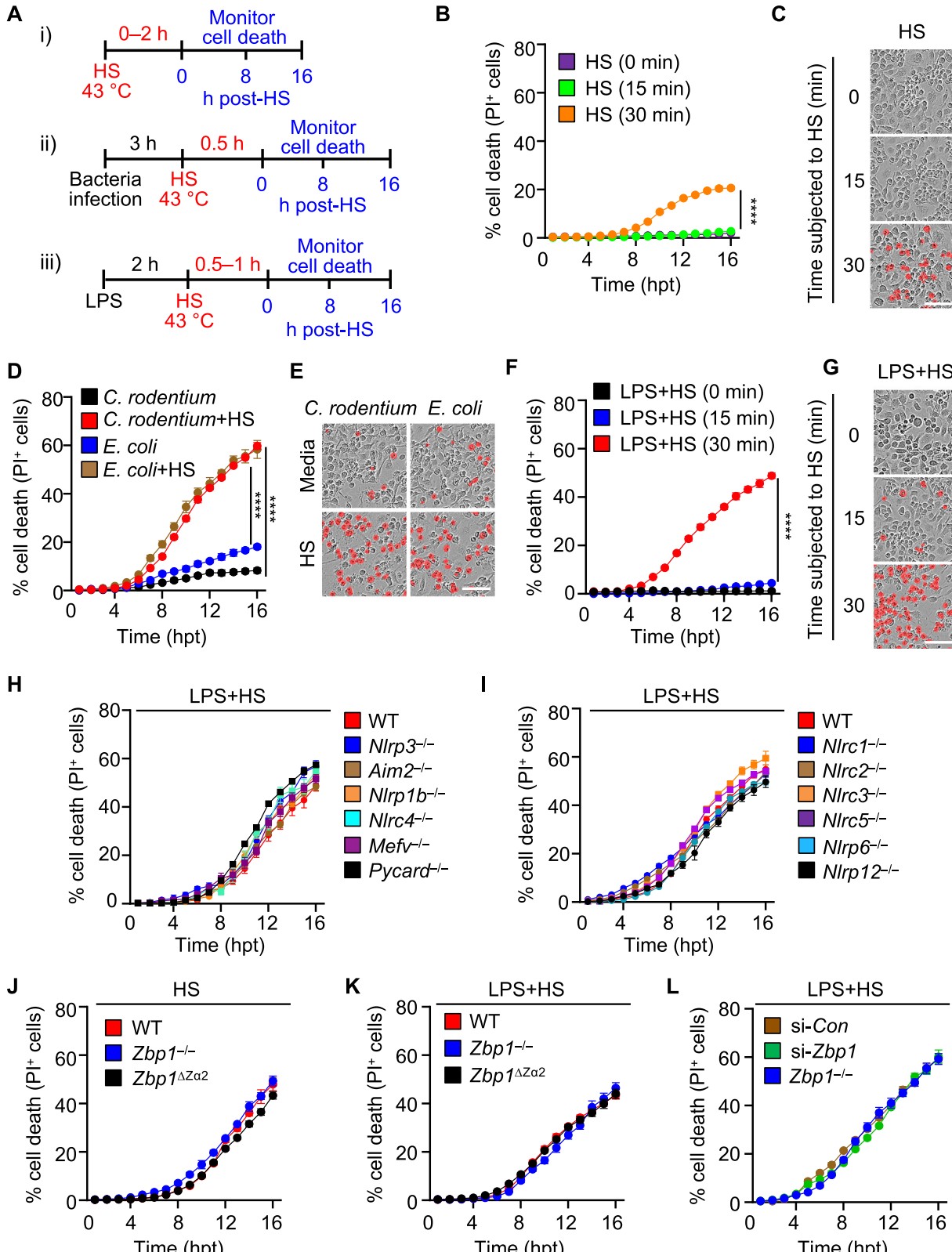

## HS triggers assembly of an NLRP3, ASC, caspase-8, and RIPK3 PANoptosome complex to induce inflammatory cell death, PANoptosis

Next, we elucidated the biochemical features of the cell death induced by HS in the presence or absence of LPS treatment. Based on our observation that LPS plus HS treatment could induce inflammasome activation and caspase-1 cleavage (Supplementary Fig. 3A), we next

determined whether HS could induce the activation of the downstream effector gasdermin D (GSDMD), which is processed by caspase-1 or caspase-11 to release a P30 pore-forming fragment[24–28]. HS alone failed to activate GSDMD (Fig. 2A). Consistent with the lack of GSDMD P30, there was no appreciable cleavage of caspase-1 or caspase-11 in response to HS alone (Fig. 2A). Another member of the gasdermin family, GSDME, can also induce inflammatory cell death under specific

**Fig. 1 | Infection potentiates heat stress-induced inflammatory cell death.**
**A** Schematic illustrating the treatment timeline for (i) heat stress (HS) alone, (ii) HS with bacterial infection, or iii) HS with lipopolysaccharide (LPS) priming. **B** Real-time analysis of cell death in bone marrow-derived macrophages (BMDMs) challenged with HS for the indicated times. **C** Representative images of cell death in (**B**) at 16 hours post-treatment (hpt). **D** Real-time analysis of cell death in *Escherichia coli* or *Citrobacter rodentium*-infected BMDMs with or without subsequent HS (43 °C for 30 min). **E** Representative images of cell death in (**D**) at 16 hpt. **F** Real-time analysis of cell death in LPS-primed BMDMs challenged with HS (43 °C for the indicated times. **G** Representative images of cell death in (**F**) at 16 hpt. **H** Real-time analysis of cell death in LPS-primed wild type (WT), *Nlrp3⁻/⁻*, *Aim2⁻/⁻*, *Nlrp1b⁻/⁻*, *Nlrc4⁻/⁻*, *Mefv⁻/⁻*, and *Pycard⁻/⁻* BMDMs following challenge with HS (43 °C for 30 min). **I** Real-time analysis of cell death in LPS-primed WT, *Nlrc1⁻/⁻*, *Nlrc2⁻/⁻*,

*Nlrc3⁻/⁻*, *Nlrc5⁻/⁻*, *Nlrp6⁻/⁻*, and *Nlrp12⁻/⁻* BMDMs following challenge with HS (43 °C for 30 min). **J** Real-time analysis of cell death in WT, *Zbp1⁻/⁻*, and *Zbp1ᐩᶻᵅ²* BMDMs following challenge with HS (43 °C for 1 h). **K** Real-time analysis of cell death in LPS-primed WT, *Zbp1⁻/⁻*, and *Zbp1ᐩᶻᵅ²* BMDMs following challenge with HS (43 °C for 30 min). **L** Real-time analysis of cell death in LPS-primed WT control siRNA (si-Con)-treated, WT *Zbp1* siRNA (si-*Zbp1*)-treated, and *Zbp1⁻/⁻* BMDMs following challenge with HS (43 °C for 30 min). **B** Data are shown as mean ± SEM; **** *P* < 0.0001 (two-tailed *t*-test; *n* = 4 from 4 biologically independent samples). **C, E, G** Images are representative of at least three independent experiments. Scale bar, 50 μm. **D, F, H−L** Data are shown as mean ± SEM; ****P < 0.0001 (one-way ANOVA with Bonferroni's multiple comparisons test; *n* = 4 from 4 biologically independent samples). Exact *P* values are presented in Supplementary data file 2.

conditions[29,30]. However, we did not observe GSDME cleavage in response to HS alone (Fig. 2A). In contrast, we found that HS alone induced the activation of caspase-8, −3, and −7 (Fig. 2B), and the phosphorylation of RIPK3 and MLKL, which were reduced over time following the removal of HS (Fig. 2C). We next compared the cell death mechanisms induced by HS alone with those induced by LPS plus HS. In line with the increased incidence of cell death in response to LPS plus HS compared with HS alone, cleavage of caspase-1, caspase-11, GSDMD, and GSDME and cleavage of caspase-8, −3, and −7 were increased and phosphorylation of RIPK3 and MLKL were sustained when LPS priming was added (Fig. 2A−C). Furthermore, recent studies have shown that activation of caspase-3 and -7 can inactivate GSDMD by processing it to produce a P20 fragment[31,32], which we also observed in response to LPS plus HS treatment (Fig. 2A). Collectively, these data suggest that LPS priming sensitizes the cells to undergo increased inflammasome activation and cell death characterized by the activation of multiple caspases and RIPKs. These results suggest that LPS plus HS induces PANoptosis, a unique, inflammatory, lytic, innate immune cell death pathway driven by caspases and RIPKs and regulated by the PANoptosome complex.

Upstream of the activation of cell death effectors, and consistent with the critical role we observed for TRIF in regulating the cell death in response to LPS plus HS, we also found that deletion of TRIF reduced the biochemical activation of the PANoptosis pathway (Supplementary Fig. 5A−C). In addition, caspase-8, which was shown to be the main regulator of apoptosis induced by HS[16], was activated independent of TRIF in response to HS alone (Supplementary Fig. 5D), while caspase-8 activation was dependent on TRIF in response to LPS plus HS (Supplementary Fig. 5B), suggesting that the mechanisms of cell death induced by HS differs with or without infection.

Since we observed that LPS plus HS induced activation of multiple cell death proteins, we hypothesized that combined inhibition or deletion of key PANoptosis components would provide complete protection against LPS plus HS-induced cell death. To test this, we treated WT BMDMs with the RIPK1 inhibitor, Nec-1, or the combination of Nec-1 with the pan-caspase inhibitor, z-VAD-FMK (zVAD). While Nec-1 treatment alone did not inhibit cell death in response to LPS plus HS compared to media control, the addition of zVAD with Nec-1 significantly inhibited the cell death (Supplementary Fig. 6A, B), indicating that RIPK1 and caspases together contribute to cell death in response to LPS plus HS. To further confirm these findings, we used a genetic model and examined *Casp1⁻/⁻Casp8⁻/⁻Ripk3⁻/⁻* (referred to as TKO) BMDMs, where major components of PANoptosis are absent. We observed reduced cell death in TKO BMDMs compared to WT BMDMs in response to LPS plus HS (Fig. 2D and Supplementary Fig. 6C). Given the activation of caspase-11 we observed in response to LPS plus HS (Fig. 2A), it is possible that the residual cell death in TKO cells may be driven by caspase-11. Indeed, deletion of caspase-11 in *Casp1⁻/⁻Casp8⁻/⁻Ripk3⁻/⁻* cells (*Casp1⁻/⁻Casp11⁻/⁻Casp8⁻/⁻Ripk3⁻/⁻*; referred to as QKO) inhibited the residual cell death observed in TKO cells stimulated with LPS plus HS (Fig. 2D and Supplementary Fig. 6C). Consistent with the

protection from cell death, we observed a reduction in GSDMD, GSDME, caspase-3, and caspase-7 cleavage in QKO BMDMs compared to WT BMDMs in response to LPS plus HS (Fig. 2E−G). Moreover, compared to WT cells, *Casp8⁻/⁻Ripk3⁻/⁻* BMDMs also showed reduced cell death (Supplementary Fig. 6D), which was further abrogated upon deletion of caspase-1 in *Casp8⁻/⁻Ripk3⁻/⁻* BMDMs (TKO) in response to LPS plus HS (Supplementary Fig. 6D). Consistently *Fadd⁻/⁻Ripk3⁻/⁻* cells phenocopied *Casp8⁻/⁻Ripk3⁻/⁻* cells (Supplementary Fig. 6D). Since caspase-1 activation was dependent on NLRP3 inflammasome activation (Supplementary Fig. 3A), we reasoned that NLRP3 likely contributed to the cell death in *Casp8⁻/⁻Ripk3⁻/⁻* BMDMs in response to LPS plus HS. Indeed, treatment of *Casp8⁻/⁻Ripk3⁻/⁻* BMDMs with the NLRP3 inhibitor MCC950 further reduced the cell death compared to *Casp8⁻/⁻Ripk3⁻/⁻* BMDMs without MCC950 (Supplementary Fig. 6E, F), suggesting that the NLRP3 inflammasome, together with RIPK3 and caspase-8, contributes to the cell death in response to LPS plus HS.

Next, we sought to understand the molecular connections among the proteins driving the cell death in response to LPS plus HS. We observed that caspase-8, the inflammasome adaptor protein ASC, and NLRP3 were all immunoprecipitated with RIPK3 endogenously in response to LPS plus HS (Supplementary Fig. 6G). Furthermore, we observed colocalization of ASC with RIPK3 and caspase-8 collectively in the same cell stimulated with LPS plus HS (Fig. 2H and I). Because ASC is known to interact with NLRP3 to assemble the inflammasome, we also assessed whether NLRP3 was in the complex. Indeed, we found that NLRP3 also colocalized with RIPK3 and caspase-8 (Supplementary Fig. 6H). Overall, these data indicate that LPS plus HS induces formation of a multi-protein PANoptosome complex consisting of NLRP3, ASC, caspase-8, and RIPK3 that collectively drive the inflammatory cell death, PANoptosis.

## Caspase-7 contributes to PANoptosis induced by LPS plus HS

Since LPS plus HS induced activation of multiple cell death molecules, including caspase-1, -3, -7, -8, -11, GSDMD, GSDME, RIPK3, and MLKL (Fig. 2A−C), we sought to understand the relative contribution of each of these molecules to the cell death using a genetic approach. Among the multiple caspases that were activated in our system and others that have been reported to regulate cell death in different contexts[33–35], *Casp1⁻/⁻*, *Casp2⁻/⁻*, *Casp3⁻/⁻*, *Casp6⁻/⁻*, *Casp11⁻/⁻*, *Casp1/11⁻/⁻*, and *Casp12⁻/⁻* BMDMs showed similar dynamics of cell death when compared with WT BMDMs (Fig. 3A, B). In contrast, *Casp7⁻/⁻* BMDMs showed reduced cell death compared to WT BMDMs (Fig. 3A, B). Since caspase-7 activation occurs downstream of caspase-8 activation, we generated *Casp7⁻/⁻Ripk3⁻/⁻* cells to determine whether caspase-7 was the main substrate downstream of caspase-8 responsible for carrying out its functions in LPS plus HS-induced cell death. We observed a similar extent of cell death in both *Casp8⁻/⁻Ripk3⁻/⁻* and *Casp7⁻/⁻Ripk3⁻/⁻* BMDMs (Fig. 3C), suggesting caspase-7 is an important executioner caspase in this cell death process.

Beyond the activation of caspases and RIPK3, we also observed activation of pore-forming executioners, including GSDMD, GSDME,

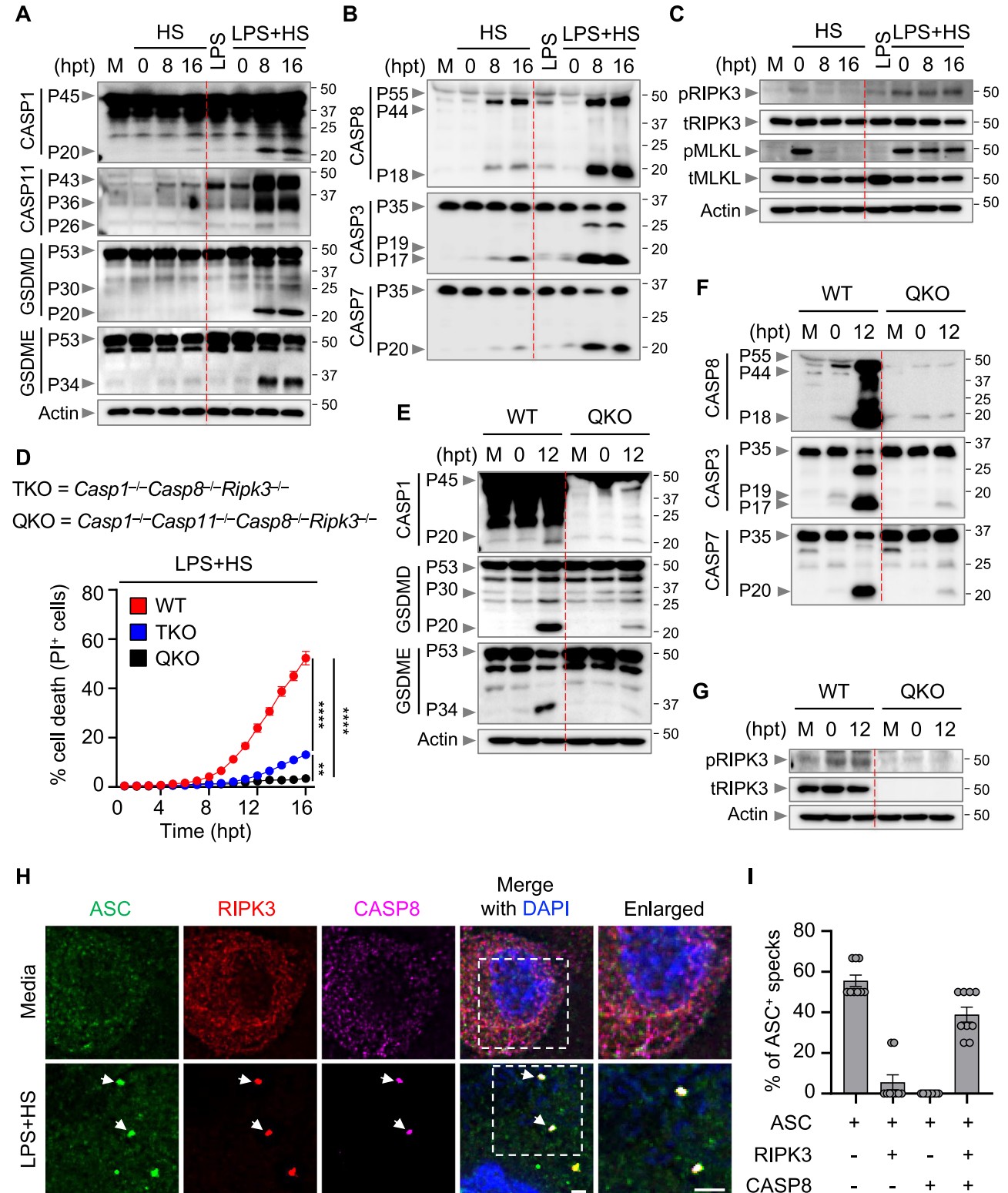

and MLKL, in response to LPS plus HS-mediated cell death (Fig. 2A–C). Caspase-1 and -11 drive pyroptosis through the cleavage of GSDMD[24–28], and caspase-8 has been reported to induce pyroptosis by cleaving GSDMD or GSDME[36,37]. In addition, RIPK3-mediated MLKL oligomerization induces MLKL pores in the membrane to execute necroptosis[38–41]. Since caspase-1, caspase-11, caspase-8, and RIPK3 together contributed to cell death in response to LPS plus HS, we sought to investigate the contribution of their canonical downstream

executioners. We found that the extent of cell death was similar in *Gsdmd*[−/−], *Gsdme*[−/−], and *Mlkl*[−/−] BMDMs with respect to WT BMDMs in response to LPS plus HS (Fig. 3D, E). MLKL-mediated necroptosis has previously been reported to regulate cell death in BMDMs during HS[16]. However, we found that *Mlkl*[−/−] BMDMs had similar cell death dynamics with respect to WT BMDMs subjected to HS in the presence or absence of LPS (Fig. 3D–G). To rule out potential redundancies among the executioner molecules, we examined *Gsdmd*[−/−]*Gsdme*[−/−], *Gsdmd*[−/−]*Mlkl*[−/−],

**Fig. 2 | Combined deletion of PANoptosis molecules inhibits cell death induced by LPS plus HS. A−C** Immunoblot analysis of **A** pro- (P45) and activated (P20) caspase-1 (CASP1); pro- (P43) and cleaved (P36 and P26) caspase-11 (CASP11); pro- (P53), activated (P30), and inactivated (P20) gasdermin D (GSDMD); pro- (P53) and activated (P34) gasdermin E (GSDME); **B** pro- (P55) and cleaved (P44 and P18) caspase-8 (CASP8); pro- (P35) and cleaved (P19 and P17) caspase-3 (CASP3); pro- (P35) and cleaved (P20) caspase-7 (CASP7); **C** phosphorylated receptor-interacting serine/threonine kinase 3 (pRIPK3), total RIPK3 (tRIPK3), phosphorylated mixed lineage kinase domain-like (pMLKL), and total MLKL (tMLKL) in wild type (WT) bone marrow-derived macrophages (BMDMs) treated with lipopolysaccharide (LPS) alone or subjected to heat stress (HS; 43 °C for 30 min) with or without LPS priming. M, media control. Actin is used as the internal control. **D** Real-time analysis of cell death in LPS-primed WT, *Casp1⁻/⁻Casp8⁻/⁻Ripk3⁻/⁻* (TKO), and *Casp1⁻/⁻Casp11⁻/⁻Casp8⁻/⁻Ripk3⁻/⁻* (QKO) BMDMs challenged with HS (43 °C for 30 min). **E−G** Immunoblot analysis of **E** pro- (P45) and activated (P20) CASP1; pro- (P53), activated (P30), and inactivated (P20) GSDMD; pro- (P53) and activated (P34)

GSDME; **F** pro- (P55) and cleaved (P44 and P18) CASP8; pro- (P35) and cleaved (P19 and P17) CASP3; pro- (P35) and cleaved (P20) CASP7; and **G** pRIPK3 and tRIPK3 in LPS-primed WT and QKO BMDMs at the indicated timepoints after HS (43 °C for 30 min). Actin is used as the internal control. **H** Immunofluorescence images of WT LPS-primed BMDMs at 12 h post-treatment (hpt) (43 °C for 30 min). Nuclei were stained with DAPI. Arrowheads indicate the colocalized ASC, RIPK3, and CASP8 specks. Images are representative of three independent experiments. **I** Distribution of ASC⁺ specks colocalized with RIPK3⁺ or/and CASP8⁺ specks in (**H**). *n* > 100 specks were counted. **A−C, E−H** Images are representative of at least three independent experiments. **D** Data are shown as mean ± SEM; **\*\*P* < 0.01, and \*\*\*\**P* < 0.0001 (one-way ANOVA with Bonferroni's multiple comparisons test; *n* = 4 from 4 biologically independent samples). **I** Data are shown as mean ± SEM (*n* = 9 from 4 biologically independent samples). **H** Scale bar, 5 µm. Exact *P* values are presented in Supplementary data file 2. For uncropped western blots, see the accompanying source data.

and *Gsdmd⁻/⁻Gsdme⁻/⁻Mlkl⁻/⁻* BMDMs and observed a similar extent of cell death in each of these genotypes with respect to WT BMDMs in response to LPS plus HS (Fig. 3D, E). Together, these results suggest that caspase-8-mediated activation of caspase-7 is a key driver of LPS plus HS-mediated PANoptosis, but that the traditional lytic executioners are not required.

### NINJ1 executes PANoptosis in response to LPS plus HS

Although combined deletion of regulators and effectors for PANoptosis protected the cells from LPS plus HS-mediated cell death (Fig. 2D−G, and Supplementary Fig. 6C, D), combined deletion of the canonical executioners failed to provide protection (Fig. 3D, E). To identify other cell death executioners and regulators involved in response to LPS plus HS, we next performed a whole-genome CRISPR-Cas9 knockout screen in murine immortalized BMDMs (iBMDMs), a well-established and widely used cell type for studying innate immune mechanisms of cell death and inflammation that can produce a nearly unlimited supply of cells for large scale analyses. After generating a pool of cells with individual genes deleted by CRISPR, we treated the cells with LPS plus HS. After 24 h, the surviving pool of cells was analyzed and compared to an untreated pool of cells to identify genes that were enriched or depleted in this treated population (Supplementary Data file 1). We then focused our analysis on the enrichment of innate immune molecules associated with programmed cell death pathways. The screen identified several gRNAs that were each significantly enriched in the surviving pool of cells (Fig. 4A), suggesting that the genes targeted by these guides played a positive role in driving the cell death in response to LPS plus HS.

Among the known cell death executioners, *Ninj1* and *Gsdmc4* were the topmost enriched genes based on *P* value (Fig. 4B). Both GSDMC and NINJ1 are known to execute inflammatory cell death. While GSDMC has recently been found to execute pyroptosis upon cleavage by caspase-8 under certain conditions[42,43], NINJ1 is known to play a central role in plasma membrane rupture during multiple forms of lytic cell death[44]. Based on their enrichment, we selected GSDMC and NINJ1 for further evaluation. We first silenced all four isoforms of *Gsdmc* in BMDMs individually to assess their roles. Silencing *Gsdmc1*, *Gsdmc2*, *Gsdmc3*, or *Gsdmc4* alone did not provide protection against LPS plus HS-mediated cell death (Supplementary Fig. 7A, B). Furthermore, we also silenced all isoforms of *Gsdmc* together to determine whether they had redundant roles, but silencing *Gsdmc1−4* also failed to provide protection from LPS plus HS-mediated cell death (Supplementary Fig. 7C, D). It is possible that other pore-forming molecules may play redundant roles with GSDMC to mediate the cell death in response to LPS plus HS. To test this, we silenced *Gsdmc1−4* in BMDMs lacking GSDMD, GSDME, and MLKL. However, similar to our results silencing *Gsdmc1−4* in WT BMDMs, we observed no reduction in cell death upon

silencing *Gsdmc1−4* in *Gsdmd⁻/⁻Gsdme⁻/⁻Mlkl⁻/⁻* BMDMs in response to LPS plus HS (Supplementary Fig. 7C, D). Together, these data suggest that GSDMC is not a required executioner, either alone or in conjunction with GSDMD, GSDME, or MLKL, for LPS plus HS-mediated cell death.

We next evaluated the role of NINJ1 in HS-mediated cell death. We found that *NINJ1* expression was upregulated in a publicly available dataset from cancer cells subjected to hyperthermia (Fig. 4C) (GSE48398[45]), providing further evidence that NINJ1 may play a role in the HS response. TLR and cytokine signaling have been shown to upregulate mRNA expression of *Ninj1* in various cells[46], and we found that NINJ1 protein expression was also increased by LPS or HS stimulation in BMDMs (Fig. 4D). To further investigate the role of NINJ1 in cell death, we used CRISPR to generate *Ninj1⁻/⁻* mice (Supplementary Fig. 8A). We then treated WT and *Ninj1⁻/⁻* BMDMs with HS in the presence or absence of LPS. We found that *Ninj1⁻/⁻* BMDMs showed reduced cell death compared with WT BMDMs (Fig. 4E−H). Glycine has been reported to block cell death by preventing NINJ1 clustering and preserving membrane integrity[47]. Indeed, we also observed that glycine treatment reduced the cell death to a similar extent that *Ninj1* deletion did in BMDMs in response to HS alone and LPS plus HS (Fig. 4E−H). In addition, glycine treatment inhibited human macrophage cell death in response to HS in the presence or absence of LPS (Fig. 4I). We also observed that the release of inflammatory DAMPs, such as HMGB1 and LDH, was reduced in *Ninj1⁻/⁻*, but not in *Gsdmd⁻/⁻* or *Gsdmd⁻/⁻Gsdme⁻/⁻Mlkl⁻/⁻*, BMDMs compared with WT in response to LPS plus HS (Supplementary Fig. 8B); however, NINJ1 deficiency did not impair the inflammasome-dependent release of IL-1β and IL-18 (Supplementary Fig. 8B), possibly due to intact activation of caspase-1 and GSDMD in *Ninj1⁻/⁻* BMDMs. Consistent with this notion, WT and *Ninj1⁻/⁻* BMDMs both showed similar activation of caspase-1, -11, GSDMD, and GSDME in response to LPS plus HS (Supplementary Fig. 8C). Furthermore, the activation of caspase-8, -3, and -7 and RIPK3 phosphorylation were not impaired in *Ninj1⁻/⁻* BMDMs (Supplementary Fig. 8D, E). Based on these results, we hypothesized that the activation of executioners such as GSDMD, GSDME, or MLKL may contribute to residual cell death in *Ninj1⁻/⁻* BMDMs. To investigate this, we treated WT, *Gsdmd⁻/⁻*, *Gsdme⁻/⁻*, *Mlkl⁻/⁻*, *Gsdmd⁻/⁻Gsdme⁻/⁻*, *Gsdmd⁻/⁻Mlkl⁻/⁻*, and *Gsdmd⁻/⁻Gsdme⁻/⁻Mlkl⁻/⁻* BMDMs with glycine to inhibit NINJ1 and assessed the cell death in response to LPS plus HS. Glycine treatment reduced the cell death in each of these genotypes to a similar level as glycine treatment in WT BMDMs (Fig. 4J). Together, these data show that NINJ1 is a key driver of cell death in response to LPS plus HS, independent of other traditional lytic cell death executioners.

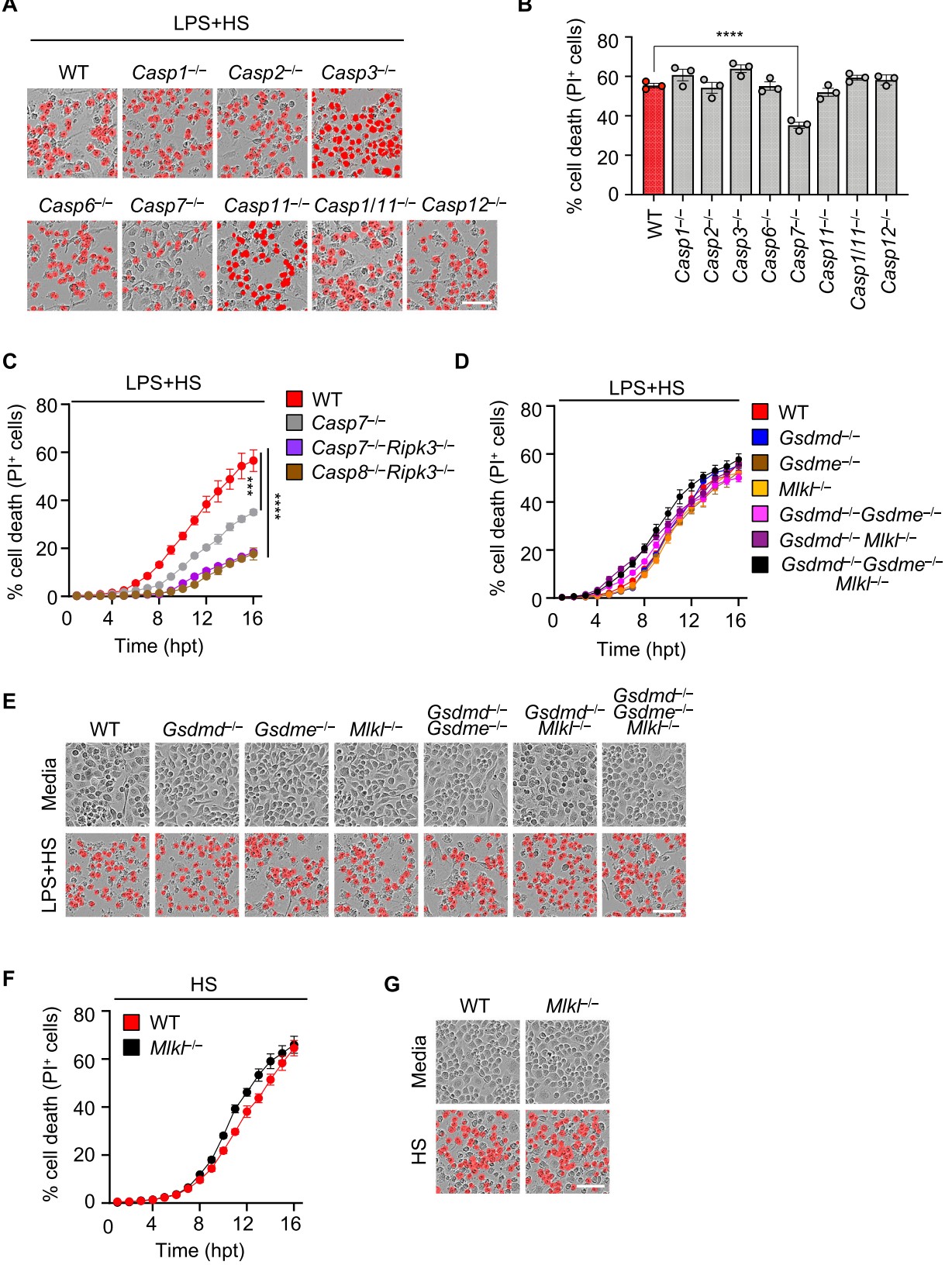

**Caspase-8 regulates NINJ1 oligomerization to drive PANoptosis in response to LPS plus HS**

While glycine treatment to inhibit NINJ1 significantly reduced cell death in WT and many executioner-deficient BMDMs in response to LPS plus HS (Figs. 4J, 5A, B), it did not provide additional protection in *Casp8⁻/⁻Ripk3⁻/⁻* or QKO (*Casp1⁻/⁻Casp11⁻/⁻Casp8⁻/⁻Ripk3⁻/⁻*) BMDMs

(Fig. 5A, B), suggesting that NINJ1 acts in the same pathway regulated by the caspase-8 axis. To test this, we first sought to understand the effect of LPS plus HS treatment on NINJ1 expression. We observed similar upregulation of NINJ1 expression in response to LPS plus HS stimulation in both WT and *Casp8⁻/⁻Ripk3⁻/⁻* BMDMs (Fig. 5C). It is known that NINJ1 undergoes oligomerization and forms speck-like

**Fig. 3 | Loss of caspase-7, but not traditional lytic cell death executioners, impairs LPS plus heat stress-induced cell death. A** Representative images of cell death in lipopolysaccharide (LPS)-primed wild type (WT), *Casp1⁻/⁻*, *Casp2⁻/⁻*, *Casp3⁻/⁻*, *Casp6⁻/⁻*, *Casp7⁻/⁻*, *Casp11⁻/⁻*, *Casp1/11⁻/⁻*, and *Casp12⁻/⁻* bone marrow-derived macrophages (BMDMs) at 16 h post-treatment (hpt) with heat stress (HS) (43 °C for 30 min). **B** Quantification of cell death in (**A**) at 16 hpt. **C** Real-time analysis of cell death in LPS-primed WT, *Casp7⁻/⁻*, *Casp7⁻/⁻Ripk3⁻/⁻*, and *Casp8⁻/⁻Ripk3⁻/⁻* BMDMs following challenge with HS (43 °C for 30 min). **D** Real-time analysis of cell death in LPS-primed WT, *Gsdmd⁻/⁻*, *Gsdme⁻/⁻*, *Mlkl⁻/⁻*, *Gsdmd⁻/⁻Gsdme⁻/⁻*, *Gsdmd⁻/⁻Mlkl⁻/⁻*, and *Gsdmd⁻/⁻Gsdme⁻/⁻Mlkl⁻/⁻* BMDMs following challenge with HS (43 °C for 30 min). **E** Representative images of cell death in (**D**) at 16 hpt. **F** Real-time analysis of cell death in WT and *Mlkl⁻/⁻* BMDMs following challenge with HS (43 °C for 60 min). **G** Representative images of cell death in (**F**) at 16 hpt. **A, E, G** Images are representative of at least three independent experiments. Scale bar, 50 µm. **B–D, F** Data are shown as mean ± SEM; *** $P < 0.001$ and **** $P < 0.0001$ (one-way ANOVA with Bonferroni's multiple comparisons test; **B** $n = 3$ from 3 biologically independent samples and **C, D, F** $n = 4$ from 4 biologically independent samples). Exact $P$ values are presented in Supplementary data file 2.

assemblies following stimulation with the NLRP3 inflammasome trigger LPS plus nigericin[44]. Consistently, we also observed oligomerization and NINJ1 speck formation in WT BMDMs treated with LPS plus HS (Fig. 5D, E). While glycine treatment inhibited NINJ1 oligomerization, speck formation, and cell death, it did not inhibit NINJ1 expression (Fig. 5D, E), suggesting that NINJ1 oligomerization and speck formation are important for inflammatory cell death in response to LPS plus HS. In addition, we found that there was reduced NINJ1 oligomerization and speck formation in *Casp8⁻/⁻Ripk3⁻/⁻* BMDMs in response to LPS plus HS (Fig. 5D, E). Together, these data suggest that caspase-8 controls NINJ1 activation by regulating its oligomerization and speck formation to drive cell death in response to LPS plus HS.

## PANoptosis through the caspase-1, -11, -8, RIPK3 axis executed by NINJ1 drives mortality

Patients experiencing HS are often also fighting infections, and these patients are highly susceptible to HS-induced mortality. Mimicking these conditions in vitro, we found that bacterial PAMPs potentiated NLRP3 inflammasome activation and inflammatory cell death in response to HS, and this cell death could be reduced by combined deletion of cell death effectors or the deletion of the executioner NINJ1 (Figs. 1–4 and Supplementary Figs. 3A and 6). To further understand the impact of this cell death in a physiological setting, we treated mice with LPS or PBS and subjected them to HS. We found that LPS plus HS treatment induced pathology in the mice (Supplementary Fig. 9), and 90% of LPS-treated mice succumbed to HS-mediated mortality, compared with only 33% of PBS-treated mice (Fig. 6A). To determine whether cell death was responsible for this mortality, we treated *Casp8⁻/⁻Ripk3⁻/⁻*, *Casp1⁻/⁻Casp8⁻/⁻Ripk3⁻/⁻* (TKO), *Casp1⁻/⁻Casp11⁻/⁻Casp8⁻/⁻Ripk3⁻/⁻* (QKO), and *Ninj1⁻/⁻* mice with LPS and subjected them to HS. *Ninj1⁻/⁻* mice had a reduced mortality rate compared with WT mice upon challenge with LPS plus HS (Fig. 6B). Moreover, pathology and mortality were also significantly reduced in mice where key PANoptosis molecules had been deleted, with improved survival in *Casp8⁻/⁻Ripk3⁻/⁻* mice and complete protection from mortality and reduced ALT and AST release in *Casp1⁻/⁻Casp8⁻/⁻Ripk3⁻/⁻* (TKO) and *Casp1⁻/⁻Casp11⁻/⁻Casp8⁻/⁻Ripk3⁻/⁻* (QKO) mice (Fig. 6B and Supplementary Fig. 9). We also observed a reduction in the release of LDH, HMGB1, IL-1β, and IL-18 in TKO and QKO mice (Fig. 6C–F), suggesting a connection between the mortality rate and the excessive release of DAMPs and cytokines induced by LPS plus HS. Together, these data suggest that inflammatory cell death, PANoptosis, mediated by caspases and RIPKs, specifically caspase-1, -11, -8, and RIPK3, and executed by NINJ1 drives mortality during HS when pathogens or PAMPs are present.

## Discussion

Despite the beneficial effects of fever during infection[5–8], HS is also associated with worse outcomes resulting in cytokine storm[9]. Our results suggest that these pathological responses to HS and LPS plus HS are driven by robust inflammasome activation and inflammatory cell death. Combined deletion of caspase-1/-11, caspase-8, and RIPK3 abolished the cell death induced by HS in the presence or absence of LPS, suggesting a critical role for PANoptosis in this context. Previous studies have shown that the execution of PANoptosis in response to

multiple infectious and inflammatory stimuli cannot be blocked by the deletion of individual molecules due to their functional redundancies, and instead requires the combined deletion of multiple cell death effectors[21,48–50]. In addition, caspase-8 is a central molecule in inflammatory cell death, and our findings suggest that caspase-8 is a key component of the LPS plus HS-induced cell death, as *Casp8⁻/⁻Ripk3⁻/⁻* BMDMs were significantly protected from cell death when compared with WT BMDMs. Downstream of caspase-8, NINJ1 and caspase-7 both partially contributed to cell death. However, *Casp1⁻/⁻Casp8⁻/⁻Ripk3⁻/⁻* BMDMs were even more protected than *Casp8⁻/⁻Ripk3⁻/⁻* BMDMs, defining functional redundancies between these molecules. Furthermore, the residual cell death observed in *Casp1⁻/⁻Casp8⁻/⁻Ripk3⁻/⁻* macrophages was further reduced upon additional loss of caspase-11, highlighting a potential role of caspase−11 in this process. In vivo, *Casp1⁻/⁻Casp8⁻/⁻Ripk3⁻/⁻* and *Casp1⁻/⁻Casp11⁻/⁻Casp8⁻/⁻Ripk3⁻/⁻* mice were fully protected from LPS plus HS-induced lethality, showing improved survival compared with both *Casp8⁻/⁻Ripk3⁻/⁻* and *Ninj1⁻/⁻* mice. These results suggest that the phenotypes we observed in BMDMs contributed to the overall pathology in mice. However, there may also be other cell death or inflammatory mechanisms involved in other cell types that contribute to the pathogenesis, and these are avenues for future study.

Some of our findings are in contrast to those of a previous study concerning the role of ZBP1 and MLKL in cell death induced by HS[16]. The previous study reported that HS-induced cell death is mediated by ZBP1 and the activation of apoptosis and necroptosis[16]. However, in our study, while we also found that molecular components involved in apoptosis and necroptosis were part of the pathway, the cell death was not driven by ZBP1 and did not rely on MLKL. The specific reason for these discrepancies remains unknown. In addition, in the context of LPS plus HS, we found that cell death was regulated by TLR signaling through TRIF. TLR signaling is known to be a key regulator of the NLRP3 inflammasome[20], and we observed NLRP3 inflammasome activation in response to LPS plus HS. In addition, non-canonical NLRP3 inflammasome activation can occur through caspase-11 sensing of LPS[51,52]. Given the role for caspase-11 in LPS plus HS-mediated cell death we observed, it is possible that extracellular LPS might gain access to the cytosol independent of NINJ1 during HS, where it can be sensed by caspase-11 to activate the inflammasome and cell death[24,53]. As an additional layer of regulation, given that several heat shock proteins (HSPs) are induced during infection and HS and are known to modulate cell death[54], HSPs may also be acting as upstream regulators of cell death molecules. These functions require further study.

Downstream of the innate immune sensors and effectors, we found that the executioner NINJ1 had a key role in the inflammatory cell death induced by HS in the presence or absence of LPS. NINJ1 oligomerization was dependent on caspase-8 and RIPK3, indicating that NINJ1 acts downstream of these molecules. NINJ1 is critical for the rupture of the plasma membrane and release of endogenous molecules such as HMGB1 and LDH in response to diverse cell death triggers[44]. Therefore, our results suggest it is possible that DAMPs released through NINJ1 pores contribute to cytokine storm and the pathological effects of HS. While previous studies have found that NINJ1 is not responsible for membrane permeability to allow the uptake of YOYO-1 or the release of IL-1β[44], others have found that PI and other molecules can pass through

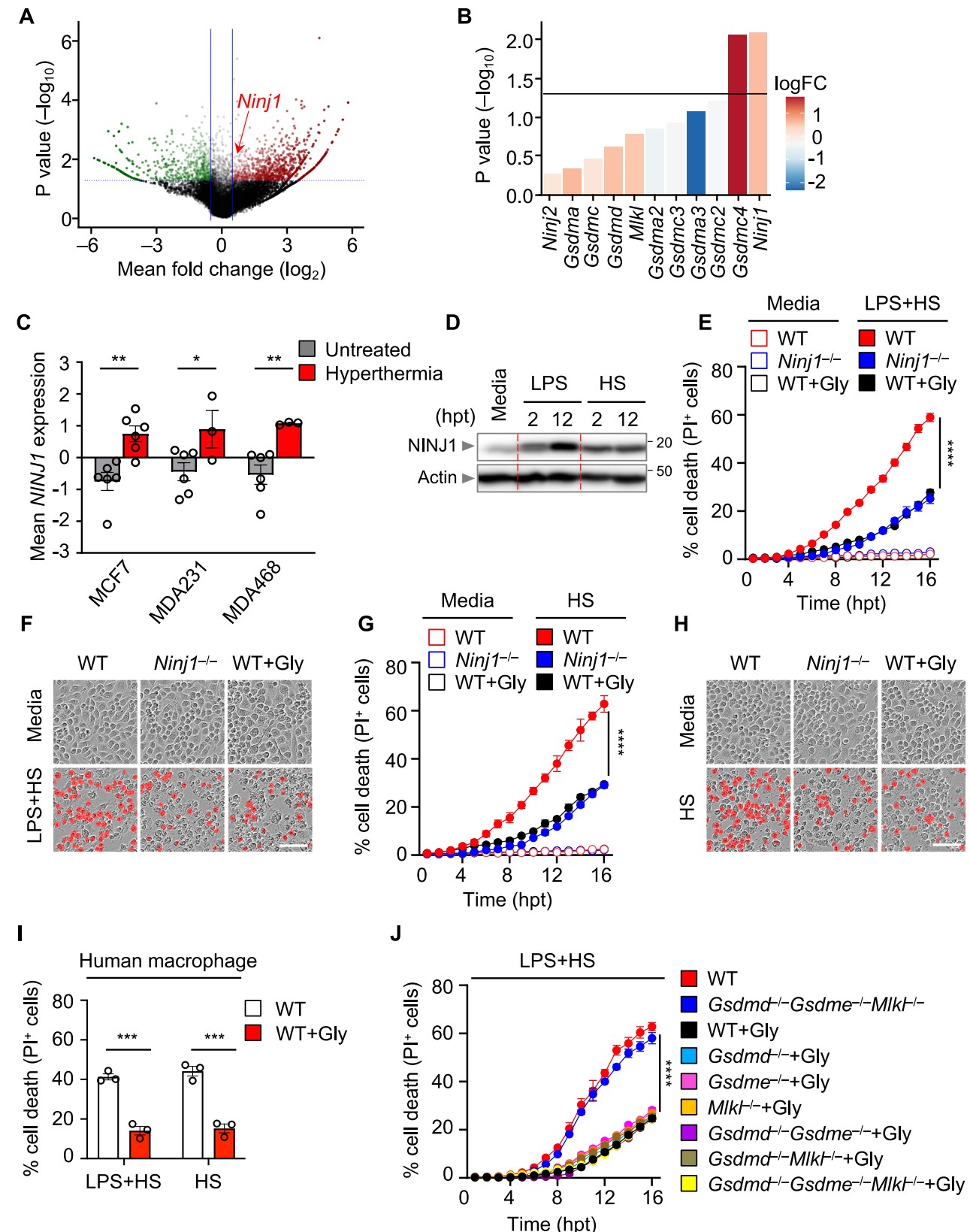

the NINJ1 pore in response to *Yersinia* infection or staurosporine treatment[55], consistent with our observations in the context of HS here. Therefore, it is possible that differences in the underlying stressors triggering cell death may account for some of the observed variations. In addition, there is a possibility that NINJ1 could influence the

oligomerization or regulatory functions of other pore-forming executioners. Thus, further studies are required to understand NINJ1 functions.

Overall, our findings suggest that the presence of multiple ligands during physiological HS conditions, including potential PAMPs,

**Fig. 4 | Genome-wide CRISPR screen identifies NINJ1 as a cell death executioner in response to LPS plus heat stress. A** Volcano plot showing the genes that are enriched or depleted in immortalized bone marrow-derived macrophages (iBMDMs) after a genome-wide CRISPR-Cas9 knockout screen of cell death induced by lipopolysaccharide (LPS) priming and heat stress (HS; 43 °C for 60 min). **B** Analysis of the enrichment of genes encoding pore-forming molecules identified in (**A**) in the CRISPR screen results. **C** *NINJ1* expression in MCF7, MDA231, and MDA468 cancer cells with and without hyperthermia (GSE48398). **D** Immunoblot analysis of NINJ1 expression in wild type (WT) BMDMs after 2 or 12 h of LPS stimulation or following HS challenge (43 °C for 60 min). **E** Real-time analysis of cell death under basal conditions (no LPS + HS) or in LPS-primed WT, *Ninj1*⁻/⁻, and glycine (Gly)-treated WT BMDMs following challenge with HS (43 °C for 30 min). **F** Representative images of cell death in (**E**) at 16 h post-treatment (hpt). **G** Real-time analysis of cell death under basal conditions (no HS) or in WT, *Ninj1*⁻/⁻, and Gly-treated WT BMDMs following challenge with HS (43 °C for 60 min). **H** Representative images of cell death in (**G**) at 16 hpt. **I** Analysis of cell

death in human macrophages 5 h after LPS + HS (43 °C for 60 min) treatment or HS (43 °C for 2 h) treatment in the presence or absence of Gly. **J** Real-time analysis of cell death in LPS-primed WT and *Gsdmd*⁻/⁻*Gsdme*⁻/⁻*Mlkl*⁻/⁻ BMDMs and Gly-treated WT, *Gsdmd*⁻/⁻, *Gsdme*⁻/⁻, *Mlkl*⁻/⁻, *Gsdmd*⁻/⁻*Gsdme*⁻/⁻, *Gsdmd*⁻/⁻*Mlkl*⁻/⁻, and *Gsdmd*⁻/⁻*Gsdme*⁻/⁻*Mlkl*⁻/⁻ BMDMs following challenge with HS (43 °C for 30 min). **B** Data are shown as mean. The detailed data analysis used for panels **A** and **B** is described in the "Methods". **C** Data are shown as mean ± SEM; *$P < 0.05$, **$P < 0.01$ (two-tailed *t*-test; $n = 3–6$ samples from GSE48398). **D, F, H** Images are representative of at least three independent experiments. **F, H** Scale bar, 50 μm. **E, G, J** Data are shown as mean ± SEM; **** $P < 0.0001$ (one-way ANOVA with Bonferroni's multiple comparisons test; $n = 4$ from 4 biologically independent samples). **I** Data are shown as mean ± SEM; ***$P < 0.001$ (two-tailed *t*-test; $n = 3$ from 3 biologically independent samples). Exact *P* values are presented in Supplementary data file 2. For uncropped western blots, see the accompanying source data.

pyrogens, cytokines, and other DAMPs, leads to the activation of inflammatory cell death, PANoptosis. Furthermore, targeting NINJ1 or other inflammatory cell death molecules has therapeutic potential in the treatment of HS-mediated pathologies. Understanding the fundamental mechanisms engaged by innate immunity to induce inflammatory responses during HS can inform the development of effective therapeutic strategies for the treatment of patients with HS and other cytokine storm-associated diseases.

## Methods

### Ethical statement
All studies involving mice or mouse tissue were performed in accordance with protocols approved by the St. Jude Children's Research Hospital committee on the Use and Care of Animals (protocol 482). Studies involving human cells were conducted using anonymous healthy blood from donors from the blood bank at St. Jude Children's Research Hospital obtained through approval of the St. Jude Institutional Review Board (IRB). Donors signed a standard informed consent document that explains the apheresis process and important details prior to donating.

### Mice
*Nlrp3*⁻/⁻ (ref. [56]), *Aim2*⁻/⁻ (ref. [57]), *Nlrp1b*⁻/⁻ (ref. [58]), *Nlrc4*⁻/⁻ (ref. [59]), *Mefv*⁻/⁻ (ref. [60]), *Nlrc1*⁻/⁻ (ref. [61]), *Nlrc2*⁻/⁻ (ref. [62]), *Nlrc3*⁻/⁻ (ref. [63]), *Nlrc5*⁻/⁻ (ref. [64]), *Nlrp6*⁻/⁻ (ref. [65]), *Nlrp12*⁻/⁻ (ref. [66]), *Cas9-GFP* knock-in (Jackson Laboratory, 026179), *Casp1*⁻/⁻*Casp11*⁻/⁻*Casp8*⁻/⁻*Ripk3*⁻/⁻ (ref. [67]), *Casp1*⁻/⁻*Casp8*⁻/⁻*Ripk3*⁻/⁻ (ref. [68]), *Pycard*⁻/⁻ (ref. [69]), *Trif*⁻/⁻ (ref. [70]), *Myd88*⁻/⁻ (ref. [71]), *Casp1*⁻/⁻ (ref. [72]), *Casp2*⁻/⁻ (ref. [73]), *Casp6*⁻/⁻ (Jackson Laboratory, 006236 (ref. [34])), *Casp11*⁻/⁻ (ref. [74]), *Casp1*⁻/⁻*Casp11*⁻/⁻ (ref. [74]), *Casp3*⁻/⁻ (ref. [75]), *Casp7*⁻/⁻ (ref. [76]), *Casp12*⁻/⁻ (ref. [77]), *Casp8*⁻/⁻*Ripk3*⁻/⁻ (ref. [78]), *Fadd*⁻/⁻*Ripk3*⁻/⁻ (ref. [79]), *Ripk3*⁻/⁻ (ref. [80]), *Gsdmd*⁻/⁻ (ref. [81]), *Gsdme*⁻/⁻ (ref. [82]), *Mlkl*⁻/⁻ (ref. [83]), *Gsdmd*⁻/⁻*Gsdme*⁻/⁻ (ref. [48]), *Gsdmd*⁻/⁻*Mlkl*⁻/⁻ (ref. [84]), *Gsdmd*⁻/⁻*Gsdme*⁻/⁻*Mlkl*⁻/⁻ (ref. [48]), *Zbp1*⁻/⁻ (ref. [85]), *Zbp1*⁻/⁻*Mefv*⁻/⁻ (ref. [23]), and *Zbp1*^ΔZα2 (ref. [86]) mice have been previously described. *Trif*⁻/⁻*Myd88*⁻/⁻ and *Casp7*⁻/⁻*Ripk3*⁻/⁻ mice were created for this study by crossing *Trif*⁻/⁻ (ref. [70]) with *Myd88*⁻/⁻ (ref. [71]) and crossing *Casp7*⁻/⁻ (ref. [76]) and *Ripk3*⁻/⁻ (ref. [80]) mice. *Ninj1*⁻/⁻ and *Zbp1*^TDK mice were newly generated using the CRISPR methods described below. WT C57/BL6 (J substrain) mice were originally obtained from Jackson Laboratory (000664) and then interbred in our colony at St. Jude. All genetically modified mouse lines were extensively backcrossed to our WT line. All mice were bred at the Animal Resources Center at St. Jude Children's Research Hospital under specific pathogen-free conditions. Both male and female age- and sex-matched 6- to 9-week-old mice were used in this study. Mice were maintained in 20–23.3 °C and 30–70% humidity with a 12 h light/dark cycle with free access to food and water at all times, unless otherwise specified in the in vivo heat shock model (described

below), and mice were fed standard chow. For all studies involving mice, mice were euthanized by $CO_2$ inhalation, followed by cervical disolcation or another approved secondary form of euthanasia. Animal studies were conducted under protocols approved by the St. Jude Children's Research Hospital committee on the Use and Care of Animals (protocol 482).

### Generation of *Zbp1*^TDK and *Ninj1*⁻/⁻ mice
The new *Zbp1* KO (*Zbp1*^TDK) and *Ninj1* KO (*Ninj1*⁻/⁻) mice were generated using CRISPR-Cas9 technology with the help of CAGE (Center for Advanced Genome Engineering) at St. Jude Children's Research Hospital. For generation of *Zbp1*^TDK mice, pronuclear-staged C57BL/6J zygotes were injected with Cas9 protein combined with the guide RNAs matching exon 2: AGTCCTTTACCGCCTGAAGA**AGG**, (pam sequence underlined). The injected zygotes were surgically transplanted into the oviducts of pseudo-pregnant CD1 females, and newborn mice carrying the desired deletion in the *Zbp1* allele were identified by targeted next-generation sequencing (NGS) and Sanger sequencing. Western blot analysis was used to confirm the loss of ZBP1 protein expression.

For generation of *Ninj1*⁻/⁻ mice, pronuclear-staged C57BL/6 J zygotes were injected with Cas9 protein combined with two guide RNAs, sgRNA1 for the 5' of exon 1: CCTGTGTTGAGTTACCCTGAA**GG**, and sgRNA2 for the 3' of exon 1: CCCTAAAGGAACTCAGCCCG**GGG** (pam sequences are underlined), to generate a long deletion of 1621 bp including exon 1 of the *Ninj1* gene. The injected zygotes were surgically transplanted into the oviducts of pseudo-pregnant CD1 females, and newborn mice carrying the desired deletion in the *Ninj1* allele were identified by NGS analysis and Sanger sequencing. The loss of protein expression was confirmed by western blot analyses.

### Cell culture
Primary mouse bone marrow-derived macrophages (BMDMs) were generated from the bone marrow of WT and the indicated mutant mice. Cells were grown for 5–6 days in IMDM (Thermo Fisher Scientific, 12440-053) supplemented with 1% non-essential amino acids (Thermo Fisher Scientific, 11140-050), 10% heat-inactivated fetal bovine serum (HI-FBS; Biowest, S1620), 30% L929 conditioned media, and 1% penicillin and streptomycin (Thermo Fisher Scientific, 15070-063). BMDMs were then seeded into antibiotic-free media and incubated overnight before use. THP-1 cells (ATCC, TIB-202) were cultured in RPMI media (Corning, 10-040-CV) supplemented with 10% HI-FBS and 1% penicillin and streptomycin and differentiated into macrophages in RPMI 1640 medium containing 20% HI-FBS and 100 ng/ml phorbol 12-myristate 13-acetate (PMA) for 2 days. The mouse macrophagic cell line RAW264.7 (ATCC, TIB-71) and fibroblast cell line L929 (ATCC, CCL-1) were cultured in DMEM (Gibco, 11995-065) supplemented with 1% penicillin and streptomycin and 10% HI-FBS. Immortalized BMDMs

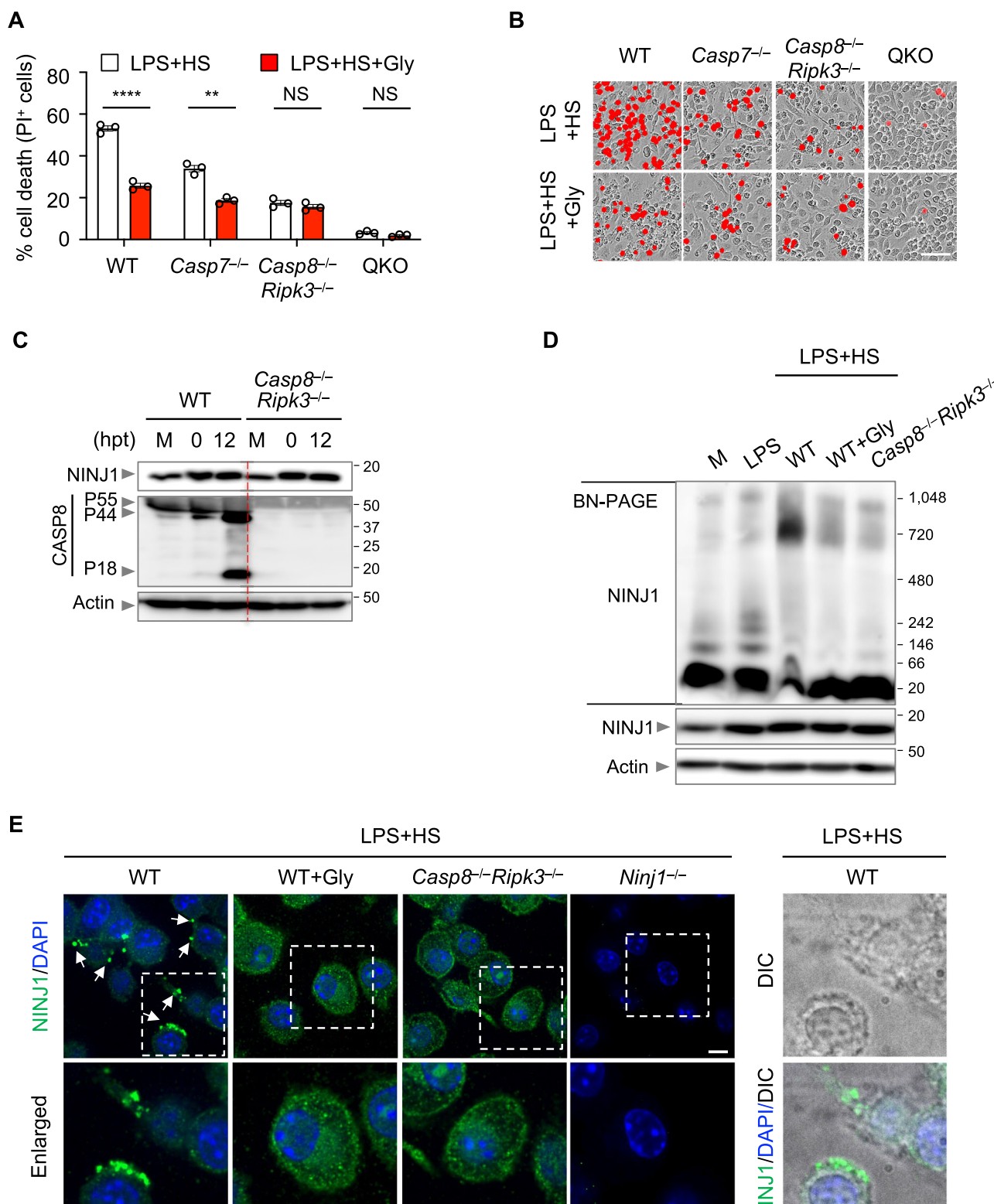

**Fig. 5 | The caspase-8/RIPK3 axis regulates NINJ1 oligomerization to control cell death. A** Analysis of cell death in lipopolysaccharide (LPS)-primed wild type (WT), *Casp7*⁻/⁻, *Casp8*⁻/⁻*Ripk3*⁻/⁻, and *Casp1*⁻/⁻*Casp11*⁻/⁻*Casp8*⁻/⁻*Ripk3*⁻/⁻ (QKO) bone marrow-derived macrophages (BMDMs) 16 h post-treatment (hpt) with heat stress (HS; 43 °C for 30 min) in the presence or absence of glycine (Gly). **B** Representative images of cell death in (**A**) at 16 hpt. **C** Immunoblot analysis of NINJ1 and pro-(P55) and cleaved (P44 and P18) caspase-8 (CASP8) in LPS-primed WT and *Casp8*⁻/⁻*Ripk3*⁻/⁻ BMDMs at the indicated timepoints after HS (43 °C for 30 min). M, media control. **D** Blue native polyacrylamide gel electrophoresis (BN-PAGE) analysis of NINJ1 in WT and *Casp8*⁻/⁻*Ripk3*⁻/⁻ BMDMs 12 h after HS (43 °C for 30 min) with the indicated conditions. SDS-PAGE analysis of NINJ1 and actin is used as the internal control. **E** Immunofluorescence images of NINJ1 in LPS-primed WT, Gly-treated WT, *Casp8*⁻/⁻*Ripk3*⁻/⁻, and *Ninj1*⁻/⁻ BMDMs 12 hpt following HS (43 °C for 30 min). Nuclei were stained with DAPI. Arrowheads indicate NINJ1 specks. The differential inter-ference contrast (DIC) image depicts the same cells as shown in the WT enlarged panel. **A** Data are shown as mean ± SEM; NS, not significant, **P < 0.01, ****P < 0.0001 (two-tailed *t*-test; n = 3 from 3 biologically independent samples). **B**–**E** Images are representative of at least three independent experiments. **B** Scale bar, 50 μm. **E** Scale bar, 25 μm. Exact *P* values are presented in Supplementary data file 2. For uncropped western blots, see the accompanying source data.

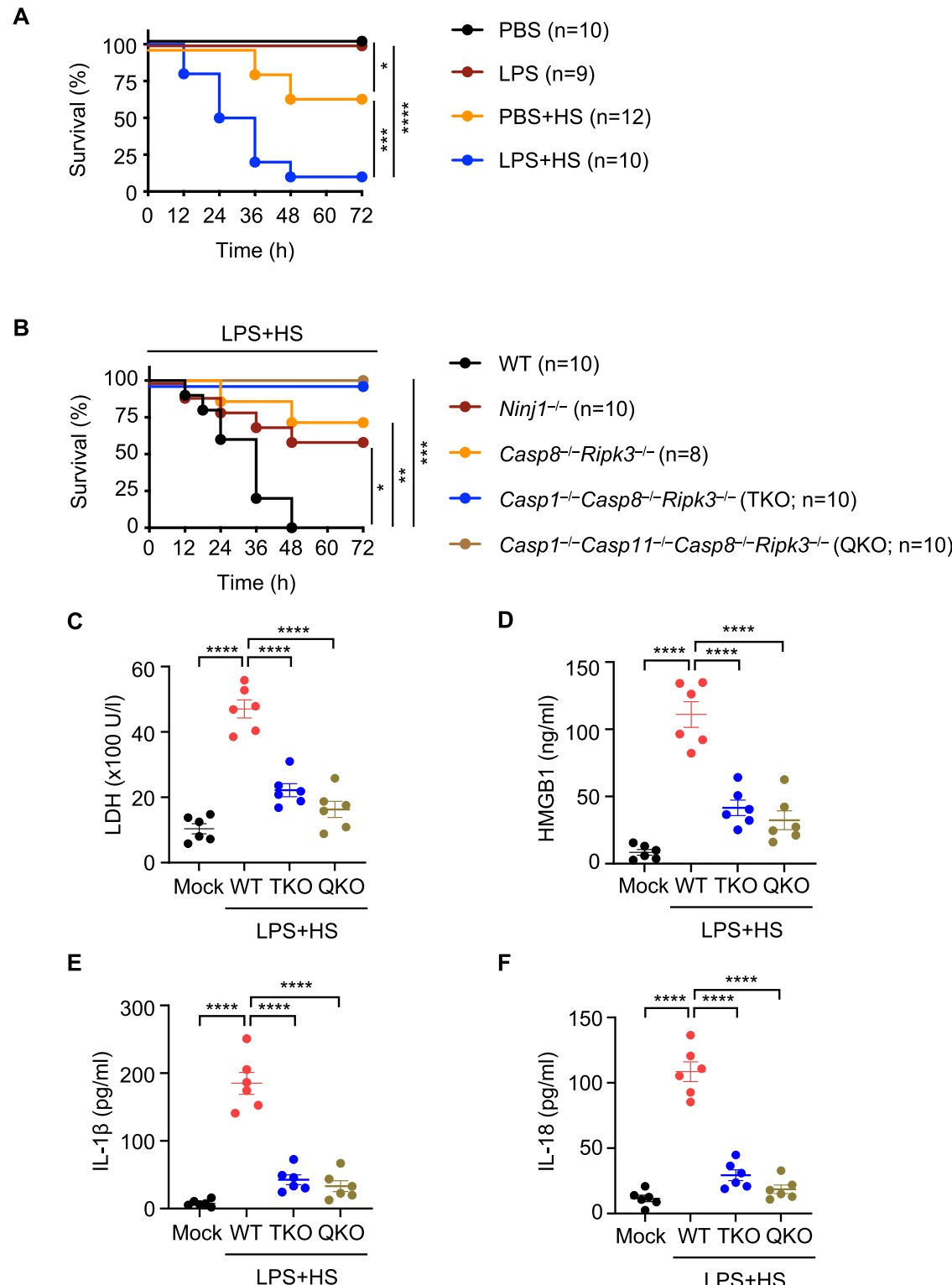

**Fig. 6 | Inhibition of inflammatory cell death, PANoptosis, provides protection against LPS plus heat stress-induced lethality and DAMP release in mice.**
**A** Survival of 6- to 8-week-old male and female wild-type (WT) mice injected with PBS or lipopolysaccharide (LPS) (5 mg/kg body weight) for 2 h, followed by heat stress (HS) at 39 °C for 2 h. The survival was monitored for 72 h following HS. PBS (n = 10), LPS (n = 9), PBS + HS (n = 12), LPS + HS (n = 10). **B** Survival of 6- to 8-week-old WT (n = 10), Ninj1$^{-/-}$ (n = 10), Casp8$^{-/-}$Ripk3$^{-/-}$ (n = 8), Casp1$^{-/-}$Casp8$^{-/-}$Ripk3$^{-/-}$ (TKO; n = 10), and Casp1$^{-/-}$Casp11$^{-/-}$Casp8$^{-/-}$Ripk3$^{-/-}$ (QKO; n = 10) mice injected with LPS (5 mg/kg body weight) for 2 h, followed by HS at 39 °C for 2 h. The survival was

monitored for 72 h following HS. **C–F** Analysis of (**C**) LDH, (**D**) HMGB1, (**E**) IL-1β, and (**F**) IL-18 levels in the serum of untreated WT mice (mock, n = 6) or WT (n = 6), TKO (n = 6), and QKO (n = 6) mice injected with LPS (5 mg/kg body weight) for 2 h, followed by HS at 39 °C for 2 h. The serum was collected at 36 h post-HS. **A–F** Data are pooled from three independent experiments. **A, B** *P < 0.05, **P < 0.01, ***P < 0.001, and ****P < 0.0001 (log-Rank [Mantel-Cox] test). **C–F** Data are shown as mean ± SEM; ****P < 0.0001 (one-way ANOVA with Bonferroni's multiple comparisons test). Exact P values are presented in Supplementary data file 2.

(iBMDMs) were cultured in DMEM (Gibco, 11995073) supplemented with 10% HI-FBS.

## Isolation of monocytes from human blood and differentiation to monocyte-derived macrophages

Fresh human blood was collected from the apheresis rings of anonymous healthy blood donors from the blood bank at St. Jude Children's Research Hospital obtained through approval of the St. Jude IRB. PBMCs were isolated from the freshly collected blood using lymphoprep solution (Stemcell Technologies, #07801/07811). Untouched naïve monocytes were purified from the peripheral blood mononuclear cells (PBMCs) using a monocyte isolation kit (Stemcell Technologies, #19669) strictly in accordance with the manufacturer's protocol. These purified monocytes were further differentiated into monocyte-derived macrophages by culturing them in RPMI media supplemented with 10% HI-FBS, 1% penicillin-streptomycin, and 20 ng/ml human MCSF (Peprotech, 300-25) for 6 days. After 3 days, an additional 10 ml of the same media was added to the cells. On day 6, total cells were harvested and washed three times with PBS. The total cells were resuspended in RPMI supplemented with 10% HI-FBS, 1% penicillin-streptomycin, and 20 ng/ml human MSCF. The cells were then used for further experiments.

## Bacteria culture

*Escherichia coli* (*E. coli*, American Type Culture Collection 11775) and *Citrobacter rodentium* (*C. rodentium*, American Type Culture Collection 51459) were grown in Luria-Bertani (LB) broth (MP Biomedicals, 3002-031) under aerobic conditions at 37 °C overnight. Bacteria were then subcultured separately at a ratio of 1:10 at 37 °C for 3 h in fresh LB broth to generate log-phase grown bacteria.

## Cell stimulation and infection

Primary mouse BMDMs, primary human macrophages, primary human monocytes, THP-1 cells, RAW264.7 cells, and L929 cells were stimulated with the following ligands where indicated unless otherwise noted: 100 ng/ml LPS (InvivoGen, tlrl-3pelps), 1 µg/ml Pam3CSK4 (Pam3; InvivoGen, tlrl-pms), 1 µg/ml poly(I:C) (InvivoGen, tlrl-picw), 1 µg/ml Imiquimod (InvivoGen, tlrl-imqs), 1 µg/ml CpG (InvivoGen, tlrl-1585), 25 µM zVAD, 30 µM Nec-1S, 1 µM MCC950, 5 µM glycine (mouse BMDMs) and 50 µM glycine (human macrophages) (Sigma-Aldrich, G7126) for 2 h. Then, cells were placed in an incubator at 43 °C and 5% $CO_2$ for indicated times to induce heat stress (HS). After HS exposure, cells were incubated at 37 °C for the indicated times. Cell lysates and supernatants were collected at the indicated timepoints after HS for western blot and immunoprecipitation.

For bacterial infection, the BMDMs were infected separately with the following bacteria: 5 MOI of *E. coli* or 5 MOI of *C. rodentium* for 3 h. Then, cells were washed three times with PBS, and fresh media supplemented with 50 µg/ml gentamicin (Thermo Fisher Scientific, 15750-060) was added to kill the extracellular bacteria. Then the cells were subjected to HS as described above.

## Real-time imaging for cell death

The kinetics of cell death were determined using the IncuCyte S3 (Sartorius) live-cell automated system. BMDMs ($1 \times 10^6$ cells/well), human macrophages ($5 \times 10^5$ cells/well), human monocytes ($5 \times 10^5$ cells/well), THP-1 cells ($5 \times 10^5$ cells/well), RAW264.7 cells ($2 \times 10^5$ cells/well), or L929 cells ($1 \times 10^6$ cells/well) were seeded in 12-well tissue culture plates. Cells treated with LPS (100 ng/ml) for 2 h or infected with bacteria for 3 h were incubated at 43 °C for indicated times, and stained with propidium iodide (PI; Life Technologies, P3566) following the manufacturer's protocol. The plate was then placed in the IncuCyte and scanned, and fluorescent and phase-contrast images (4 image fields/well) were acquired in real-time every 0.5 or 1 h from 0 h post-treatment at 37 °C and 5% $CO_2$. PI-positive dead cells were marked with

a red mask for visualization. The image analysis, masking, and quantification of dead cells ($PI^+$ cells) were done using the software package supplied with the IncuCyte imager.

## Immunoblot analysis

Immunoblotting was performed as described previously[87]. In brief, for caspase analysis, cell lysates and culture supernatants were combined in 50 µl caspase lysis buffer (containing 1× protease inhibitors, 1× phosphatase inhibitors, 10% NP-40, and 25 mM DTT) and 4× sample loading buffer (containing SDS and 2-mercaptoethanol). For immunoblot analysis of signaling components, cytokines, and DAMPs, supernatants were removed, and cells were washed once with DPBS (Thermo Fisher Scientific, 14190-250), followed by lysis in RIPA buffer and sample loading buffer. For immunoblot analysis of LDH, HMGB1, mature IL-1β, and IL-18 in the supernatant, the supernatant was collected and centrifuged at 8000 × *g* for 2 min. After removing cell debris, the obtained supernatant was combined in 4× sample loading buffer. Proteins were separated by electrophoresis through 8–12% polyacrylamide gels. Following electrophoretic transfer of proteins onto PVDF membranes (Millipore, IPVH00010), nonspecific binding was blocked by incubation with 5% skim milk, then membranes were incubated with primary antibodies against: anti-caspase-1 (AdipoGen, AG-20B-0044, 1:1000), anti-caspase-11 (Novus Biologicals, NB120-10454, 1:1000), anti-GSDMD (Abcam, ab209845, 1:1000), anti-GSDME (Abcam, ab215191, 1:1000), anti-caspase-8 (Cell Signaling Technologies [CST], 4927, 1:1000), anti-cleaved caspase-8 (CST, 8592, 1:1000), anti-caspase-3 (CST, 9662, 1:1000), anti-cleaved caspase-3 (CST, 9661, 1:1000), anti-caspase-7 (CST, 9492, 1:1000), anti-cleaved caspase-7 (CST, 9491, 1:1000), anti-pRIPK3 (CST, 91702 S, 1:1000), anti-RIPK3 (ProSci, 2283, 1:1000), anti-pMLKL (CST, 37333, 1:1000), anti-MLKL (Abgent, AP14272b, 1:1000), anti-NLRP3 (AdipoGen, AG-20B-0044, 1:1000), anti-ZBP1 (AdipoGen, AG-20B-0010, 1:1000), anti-NINJ1 (1:1000) (ref. [88]), anti-LDHA (Proteintech, 19987-1-AP, 1:1000), anti-HMGB1 (Abcam, ab79823, 1:1000), anti-cleaved IL-1β (CST, 63124, 1:1000), anti-IL-1β (CST, 12426, 1:1000), anti-IL-18 (Abcam, ab207323, 1:1000), anti-ASC (Adipogen, AG-25B-0006, 1:1000), anti-GAPDH (Santacruz, sc-166574 HRP, 1:10,000), and anti-β-actin (Proteintech, 66009-1-IG, 1:5000). Membranes were then washed and incubated with the appropriate horseradish peroxidase (HRP)–conjugated secondary antibodies (Jackson ImmunoResearch Laboratories, anti-rabbit [111-035-047], 1:5000; anti-mouse [315-035-047], 1:5,000; and anti-rat [112-035-003], 1:5000) for 1 h. Proteins were visualized using Immobilon Forte Western HRP Substrate (Millipore, WBLUF0500), and membranes were developed with the GE Amersham Imager 600. Images were analyzed with ImageJ (v1.53a).

## Immunofluorescence staining

BMDMs were seeded on coverslips at $5 \times 10^5$ cells/well and placed in 24-well plates and stimulated as described above. After the stimulation was completed, cells were fixed in 4% paraformaldehyde for 15 min at room temperature, followed by permeabilization for 3 min in 0.5% Triton X-100. Then, cells were blocked in 10% normal goat serum (Life Technologies, 01-6201) for 1 h at room temperature. Samples were incubated with anti-ASC (Millipore, 04-147, 1:100), anti-NLRP3 (AdipoGen, AG-20B-0044, 1:200), anti-RIPK3 (Millipore, MABC1595, 1:200), anti-caspase-8 (CST, 4927, 1:200), or anti-NINJ1 (1:200) (ref. [88]) overnight at 4 °C. Cells were then washed three times with PBS and incubated with Alexa Fluor 488-conjugated antibody against mouse immunoglobulin G (IgG) (Thermo Fisher Scientific, A11029, 1:200), Alexa Fluor 568-conjugated antibody against rat IgG (Thermo Fisher Scientific, A11077, 1:200), or Alexa Fluor 647-conjugated antibody against rabbit IgG (Thermo Fisher Scientific, A21245, 1:200) for 2.5 h at room temperature. Then, cell nuclei were counterstained by 4',6-diamidino-2-phenylindole (DAPI) and observed and imaged using the Marianas spinning disc confocal

system (Intelligent Imaging Innovations) comprised of an inverted AxioObserver Z.1 microscope (Carl Zeiss), CSU-W1 with SoRa (Yokogawa), Prime95B sCMOS camera (Photometrics), 405 nm, 473 nm, 561 nm, 647 nm solid state laser lines (Coherent), and a 1.4NA 100× oil objective. Images were acquired using Slidebook 6 software with the laser power set as current and the exposure time at 200 ms for each channel. The quantification of PANoptosome complexes was done with images obtained with the 60× objective and counted manually.

## Immunoprecipitation assay

The immunoprecipitation assay was performed as described previously[34]. Primary mouse BMDMs seeded into 10-cm dishes were primed with LPS for 2 h and subjected to HS (43 °C for 30 min). After HS, the cells were incubated at 37 °C for 12 h. After washing with DPBS, cells were lysed in Triton X-100 lysis buffer (20 mM Tris, pH 7.4, 137 mM NaCl, 2 mM EDTA, 1% Triton X-100, 10% glycerol, 1 × complete protease Inhibitor). Supernatant was collected, and 500 µg of cell lysate was incubated with 1 µg of anti-RIPK3 antibody (ProSci, 2283) at 4 °C overnight. The lysate immunoprecipitated with anti-IgG (CST, 3900) served as a negative control. The immune complexes were then purified with 30 µl of protein A magnetic beads (Millipore, LSKMAGA02) at 4 °C for 4 h, then centrifuged and washed with lysis buffer. The immunoprecipitated proteins were further analyzed by western blotting.

## BN-PAGE

Primary mouse BMDMs were lysed with native-PAGE lysis buffer (150 mM NaCl, 1% Digitonin, 50 mM Tris pH 7.5, and 1 × complete protease inhibitor). After centrifuging at 20,800 × g for 30 min, lysates were mixed with 4 × NativePAGE sample buffer and Coomassie G-250 (Thermo Fisher Scientific, BN2008). Then, samples were subjected to BN-PAGE using NativePAGE 3–12% Gel (Thermo Fisher Scientific, BN1001BOX) and NativeMark Unstained Protein Ladder (Thermo Fisher Scientific, LC0725).

## siRNA-mediated gene silencing

A total of 5 nmol siRNA was dissolved in sterile nuclease-free water to a final concentration of 50 µM, and 0.5 µl siRNA was added to $1 \times 10^6$ BMDMs. Electroporation was performed using the neon transfection system (Invitrogen), with parameters −1500 V, 1 pulse and 20 ms width. After electroporation, BMDMs were immediately transferred into 12-well plates with a seeding density of $1 \times 10^6$ cells per well. After 48 h of transfection, BMDMs were stimulated with LPS-priming and HS (43 °C for 30 min) to assess cell death. All siRNAs were purchased from Horizon Discovery with the following catalog numbers: Gsdmc1: M-050512-01-0005, Gsdmc2: M-053892-01-0005, Gsdmc3: M-054224-01-0005, Gsdmc4: M-049449-01-0005, Zbp1: M-048021-00-0005, and control: D-001206-13-05.

## Generation of whole-genome iBMDM-Brie lentiviral library

The Mouse Brie CRISPR KO library carrying four gRNAs for each gene to cover the entire genome was a gift from David Root and John Doench (Addgene #73632 and #73633). The plasmid library was amplified and validated in the Center for Advanced Genome Engineering (CAGE) at St. Jude as described in the Broad GPP protocol, the only exception being the use of Endura DUOs electrocompetent E. coli cells. The St. Jude Hartwell Center Genome Sequencing Facility provided all NGS sequencing. Single end 100 cycle sequencing was performed on a NovaSeq 6000 (Illumina). Validation to check gRNA presence and representation was performed using calc_auc_v1.1.py (https://github.com/mhegde/) and count_spacers.py[89]. Viral particles were produced by the St. Jude Vector Development and Production laboratory using co-transfection of the lentiviral packaging plasmids along with the Brie library in the 293T cells. CRISPR KO screens were analyzed using Mageck-Vispr v0.5.7 (ref. 90).

## Genome-wide CRISPR-Cas9 screen

Cas9-expressing iBMDMs were generated from Cas9-GFP knock-in mice. A total of $300 \times 10^6$ Cas9-iBMDMs were distributed across twelve 15 cm$^2$ tissue culture dishes at $25 \times 10^6$ cells per dish and infected with the Brie library of lentiviral particles, which carry four gRNAs for each gene to cover the entire genome, at an MOI of 0.3 in 25 ml of complete DMEM (DMEM supplemented with 10% HI-FBS with 100 ml of Lenti-BOOST transduction reagent (Siron Biotech, SB-P-LV-101-02, Lentivirus Transduction Enhancer Solution). After infection, the iBMDMs were incubated for 24 h for efficient transduction. These transduced cells were expanded with intermittent passaging to avoid overcrowding of the cells and to generate a sufficient number of cells for the downstream whole-genome CRISPR screens. Two replicates of an adequate number of cells were used as control to obtain a representation (screen depth) of >500 cells for each sgRNA of the library, and a similar number of cells from the same batch of virus preparation were stimulated with LPS priming for 2 h, then incubated at 43 °C for 1 h to induce HS. After 24 h, the media was removed, and cells in the unstimulated and stimulated conditions were washed with PBS to remove the non-adherent dead cells, leaving only the adherent surviving cells for downstream analyses. The surviving cells from the unstimulated and LPS plus HS-treated samples were then subjected to CRISPR screen enrichment analysis. Total genomic DNA was isolated using NucleoSpin Blood kits (Takara Bio Inc., USA; 740954 and 740950) and the concentrations of the isolated gDNA samples were measured using NanoDrop (Thermo Fisher Scientific, USA).

Next-generation sequencing (NGS) of the PCR-amplified, barcoded gRNAs was performed to quantitatively identify gRNAs that were enriched in the surviving pool of cells. Because the presence of a gRNA should delete the corresponding gene in that cell, enriched gRNAs were expected to represent genes where deletion rescues the cells from LPS plus HS-mediated cell death, suggesting they are positive regulators of the cell death.

Using the MAGeCK pipeline, the $\log_2$fold change was estimated with significance levels for the genes in the CRISPR screen. The genes with positive fold change were expected to be important for cell death. The topmost enriched genes along with their significance from the CRISPR screen were highlighted using a volcano plot using MAGeCK-Flute v2.0.0 (ref. 91).

## Microarray analysis

Publicly available data were collected from GEO, accession id GSE48398 (ref. 45). The dataset consisted of three malignant breast cancer cell lines (MDA231, MDA468, and MCF7) subjected to hyperthermic shock and demonstrating enhanced sensitivity. Quality control steps were performed, including normalized quantiles using the 'normalize.quantiles' function from preprocessCore v1.58.0 package when the counts are not normalized, followed by $\log_2$ transformation for downstream differential expression analysis. Differential expression analysis was performed using the limma v3.52.1 (refs. 92,93) package in R v4.1.1. The P value < 0.05 was used to estimate the set of differentially expressed genes owing to the small sample size (6 controls and 3 hyperthermic shock treated breast cancer cell lines).

## In vivo HS model

Age- and sex-matched, 6- to 8-week-old WT, Ninj1$^{-/-}$, Casp8$^{-/-}$Ripk3$^{-/-}$, Casp1$^{-/-}$Casp8$^{-/-}$Ripk3$^{-/-}$, and Casp1$^{-/-}$Casp11$^{-/-}$Casp8$^{-/-}$Ripk3$^{-/-}$ mice were used to induce HS in the presence or absence of LPS treatment. Mice were injected intraperitoneally with PBS or a sub-lethal dose of LPS (5 mg/kg body weight; Sigma, L2630). To induce HS, mice treated with PBS or LPS for 2 h were placed in a heating chamber maintained at 39 °C for an additional 2 h. Mice were restricted to food and water during the HS stimulation. Following HS, the mice were subsequently

moved back to their original cages and maintained at room temperature with free access to food and water. The mice were monitored over a period of up to 3 days for survival.

## Cytokine and serum parameter measurements

In vivo cytokines and inflammatory parameters were detected in the serum by using ELISA for LDH (Promega, G1780), HMGB1 (Novus, NBP2-62767), IL-1β (Invitrogen, 88-7013), and IL-18 (Invitrogen, BMS618-3), and by using colorimetry with ALT and AST measurement kits (HORIBA, A11A01627 and A11A01629, respectively), all according to the manufacturer's instructions.

## Quantification and statistical analysis

GraphPad Prism 9.0 software was used for data analysis. Data are presented as mean ± SEM. The student's t tests (two-tailed) for two groups or ANOVA (with Bonferroni's multiple comparisons test) for three or more groups were used to determine the statistical significance. $P$ values less than 0.05 were considered statistically significant where $*P < 0.05$, $**P < 0.01$, $***P < 0.001$, and $****P < 0.0001$.

## Reporting summary

Further information on research design is available in the Nature Portfolio Reporting Summary linked to this article.

## Data availability

Next-generation sequencing results from the CRISPR screen are deposited in the Gene Expression Omnibus (GEO) database under accession code GSE252609. The publicly available dataset from cancer cells subjected to hyperthermia was re-analyzed for this study and is available in the GEO database under accession code GSE48398, originally published by Amaya, C et al. (ref. 45). All other datasets generated and analyzed in this study are provided within the manuscript and the accompanying supplementary figures, tables, and data files or from the corresponding author upon reasonable request. Source data are provided with this paper.

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

## Acknowledgements

We thank all the members of the Kanneganti laboratory for their comments and suggestions during the development of this manuscript. We thank R. Tweedell, PhD, for scientific editing and writing support, and Katie Thorne for in vivo assistance. We thank the labs of Dr. Benjamin Youngblood and Dr. Caitlin Zebley for assistance with acquiring apheresis rings. We thank Dr. Shizuo Akira for supplying the *Nlrc5⁻/⁻* mice. We thank Dr. Vishva Dixit and Dr. Nobuhiko Kayagaki (Genentech) for the *Casp1⁻/⁻Casp11⁻/⁻* and *Casp11⁻/⁻* mutant mouse strains, Deltagen for the *Casp12⁻/⁻* mutant mouse strain, and Dr. Masahiro Yamamoto for the *Trif⁻/⁻* mutant mouse strain. Work from our laboratory is supported by the US National Institutes of Health (AI101935, AI124346, AI160179, AR056296, and CA253095 to T.-D.K.) and the American Lebanese Syrian Associated Charities (to T.-D.K.). The content is solely the responsibility of the authors and does not necessarily represent the official views of the National Institutes of Health.

## Author contributions

J.H.H., R.K., and T.-D.K. conceptualized the study; J.H.H. and R.K. designed the methodology; J.H.H., R.K., R.K.S.M., R.S., and B.R.S. performed the experiments; R.M., J.K., H.B., and S.M.P.-M. conducted the analysis; H.P. and S.-J.B. contributed reagents and key experimental protocols; J.H.H. and R.K. wrote the manuscript with input from all authors; T.-D.K. acquired the funding and provided overall supervision.

## Competing interests

T.-D.K. was a consultant for Pfizer. The remaining authors declare no competing interests.
