## [Peer Review File · Nature Communications]

NINJ1 mediates inflammatory cell death, PANoptosis, and lethality during infection conditions and heat stressEditorial Note: This manuscript has been previously reviewed at another journal that is not operating a transparent peer review scheme. This document only contains reviewer comments and rebuttal letters for versions considered at *Nature Communications*. Mentions of the other journal have been redacted.

REVIEWER COMMENTS

Reviewer #1 (Remarks to the Author):

I reviewed this manuscript for [redacted] earlier and disagreed with other reviewers on their pointed out weaknesses and perceived lack of novelty. This manuscript went through a high quality revision at [redacted], where the authors provided a large body of new experiments to further support their conclusion.

In my opinion, this is a very compelling and high-quality manuscript about the mechanism of heat stress induced cell death. The authors use genetic deletion or siRNA silencing of all possible key players involved in pyroptosis, apoptosis and necroptosis to dissect their respective contributions to- and to dissect the pathway of heat stress induced cell death revealing PANoptosis as the involved mechanism.

I have no additional questions and consider this manuscript to be of high quality, high importance to the field and an essential complementation and extension of earlier studies.

Reviewer #4 (Remarks to the Author):

The authors demonstrate a cell death response to heat stress alongside TLR signaling through TRIF, inducing cell death through Caspase8-Ripk3, Caspase-1/11, and NINJIN1. However, these findings seem to conflict with previous published findings of heat stress induced cell death. No resolution or serious discussion is given on why that may be. Yet, I am missing mechanistic depth on how this pathway is triggered. Instead, the authors focus

on searching for executioners of this cell death. The complex genetic analysis using a variety of mouse KO is appreciated, and the progression from individual regulators to triple KO and quadruple KO, providing more and more complete cell death reduction. Highlighting the complexity and redundancy of this form of cell death is interesting, but this cannot cover the shortcomings in explanation of how the signals are triggered. In addition, the authors make some conclusions from their cell death-Crispr screen that I don't really find well supported. Overall, this publication has substantial shortcomings. It is missing some essential controls, some necessary additional analyses, and shows a lack of depth on search for mechanism. An earlier publication by Yuan et al Science 2022 also discusses heat stress and this takes some novelty away, in addition to providing different results about signaling molecules – this is also difficult to comprehend.

Specific comments:

It is unsatisfying that little effort has been made to identify which mechanisms may trigger the heat related stress signaling.

I cannot consolidate Yuan, F. et al.'s finding that ZBP1 is key to HS cell death, which did not involve TRIF. With the findings here that are dependent on TRIF but do not involve ZBP1. Especially in Extended figure 4, when performing the 1hr heat stress (HS) without LPS, which appears identical to Yuan, F. et al 's HS method. Could the authors highlight differences between your methods that may explain this?

Figure 2A: HS of 30min alone failed to activate GSDMD, caspase-1/11, or GSDME, but earlier was shown that heat stress of 30min did not cause robust cell death. So this is not surprising, it should serve as a negative control. Are the authors trying to claim that HS death without LPS is not pyroptotically driven? For that claim it would be more telling if HS that does cause robust death was used, like HS induced on a longer period of time. Then it could be analyzed for what executor molecules are responsible for cell death without LPS.

Ext. Fig. 5 G-H: The authors claim this LPS/HS condition forms one complex with NLRP3, RIPK3, and Casp8. However, NLRP3 is not probed in the co-immunoprecipitation blots and

only one image of one NLRP3 speck colocalizing with Ripk3 and Caspase-8 was provided. Furthermore, in the one image provided, there are other NLRP3 specks present that do not colocalize with Ripk3 and Casp8. I'm not certain this has to be one big speck complex, as different complexes could be formed in one cell at the same time.

Fig 3 D & F: Figure 2 blots show that HS+LPS induce GsdmD, GsdmE, and MLKL activation, but then in these figures, none of them contribute to PI uptake at all. When these molecules are activated, they make pores on the membrane, making the membrane permeable to dyes such as PI. Thus, the authors conclude that another pore must be made that can supersede all of these. However, an alternative explanation could be that these activated molecules are unable to aggregate on the cell membrane and form pores in HS conditions. Has gasdermin/MLKL oligomerization and pore formation been analyzed? This could be examined.

Fig 4A: Crispr cell death screen: Much of these data are difficult to understand, including the evaluation put forward in the paper. First, I am trying to find out how the screen was done, and the various descriptions appear incomplete. Information about this needs to be clearly stated. Importantly - what is the read-out in the screen? If it is PI, then Ninj1 should not show up as it regulates larger cell ruptures and not PI permeability. Ninj1 is not a topmost enriched gene as it's ranked 5000+ in fold-change and 300+ by p-value. GsdmC isoforms are also ranked 300+ by p-value and in the thousands by fold-change. So, it doesn't appear that this whole genome CRISPR screen supports the claims made. It is then not surprising that silencing all isoforms of GsdmC failed to provide protection. I would say that the interpretations from the authors on this aspect are not supported well. I don't think it is possible to identify Ninj1 from the screen alone.

Fig 4E-J: As for Ninj1, Ninj1 is necessary for LDH and DAMP release, as well as lysis of the cell. But in Kayagaki et al. in Nature 2021 where this was discovered, Ninj1 was not responsible for membrane permeability in uptake of YOYO-1, or the release of IL-1b. Showing equivalent YOYO-1 uptake as WT cells when undergoing pyroptosis by LPS electroporation. Thus, it is surprising that in LPS+HS and HS conditions, Ninj1 deletion resulted in a significant reduction in PI uptake. I cannot understand this discrepancy and it

should be carefully discussed. It would be highly informative then to have a negative control in these PI uptake experiments, and at the very least a no treatment condition. At minimum Furthermore, it appears that Ninj1 deletion did not fully suppress cell death, though it is hard to tell without comparing to no treatment. This may imply that Ninj1 is not solely responsible for executing cell death by HS, but plays a large significant role in its membrane permeability.

Fig 5D: Why is there no ladder provided?

Fig 6: The authors portray these experiments as modeling a patient who is fighting off infection with an uncontrollable fever. However, mice lower their body temperature when they are undergoing inflammation and infection. Thus, forcing their body temperature over base body temperature while undergoing LPS shock may have vast effects on their immune system and thermoregulation across bodily systems. They inject IP LPS to induce shock (5mg/kg LPS) and place them under heat stress of 39C for 2hrs. An analysis of what this does to the body and the immune system, such as by analyzing electrolytes, cytokines, and the activity of immune cells would be informative and necessary to understand the physiology and the context. Comparing these metrics to heat stress without LPS would provide better information about the setting. Though in vivo protection does partially line up with their proposed pathway of Caspase-8-Ripk3 and Ninj1, without additional data it is hard to define if these results are equivalent to those demonstrated by BMDMs in vitro. At least the T-KO and Q-KO giving 100% survival does indicate that survival is regulated by caspases. But additional data is required to complete our understanding of what is occurring in this model.

The discussion proposes cytokine storms as a cause for the pathological effects of HS. However, the paper does little to no investigation of cytokine release and regulation. Only in Extended Fig. 7 are cytokines even measured, and only IL-1b and IL-18 in one in vitro experiment. More should be done on measuring cytokines throughout, such as by ELISA.

Extended figure 2 demonstrates that TLR3/TLR34 responses are the most robust in causing cell death under heat stress. Usually Poly(I:C) and LPS provide priming for secondary signals, but provide little cell death on their own, yet these primed, heat stressed cells are directly

going into programmed death. Heat stress can cause a wide range of effects on the cell. So it is curious why TLR3 and TLR4 respond so robustly to this heat stress induced death, but not the other TLRs tested. TRIF signaling of course is the easiest explanation, and the KOs confirm the expectation that TRIF is important for LPS+HS. Yet TRIF is early in TLR3/TLR4 signaling cascade, and it is hard to dissociate what molecules are affected by HS, and if it is signaling through the expected TRIF pathway. Could the authors address this?

Minor comments:

Perhaps testing of other TRAF related responses independent of TLRs, such as TNFR1, ILR1, or RIG1, would be informative. As well as looking at TRAF3, other TRAFs, TBK1, or IRF3.

Later the authors mention 1hr of HS generates similar cell death in BMDMs to that seen in LPS primed cells. Why not test if this cell death is also TRIF dependent?

Reviewer #5 (Remarks to the Author):

I was asked to review whether Rev #3's questions have been addressed. Upon investigating questions raised by both Rev #3 and #2, and the answers by the authors, I would like to present my analysis below.

The reviewers' questions were largely about the novelty of the study and whether the study provided sufficient advance over previously published papers to warrant its publication in **[redacted]**. I imagine that the authors then transferred their study to Nat Comm by addressing the comments. In my opinion, the study did have some overlap with previous papers, but because NINJ1 is still quite a new molecule implicated in membrane rupture post cell death, the new insights would still serve the field well. The publication in a lower tier, but still very good, journal of this study would further expand the interest of the field in this molecule. In particular, the conclusions that heat stress activates panoptosis in addition to apoptosis and necroptosis and that NINJ1 is involved in this process are useful to the field. I also think that the in vivo data on NINJ1 KO adds additional insights to the current understanding. Thus, I support the publication of the data.

We are grateful for the comments from Reviewer 1 that this is a “very compelling and high-quality manuscript”, and that is of “high importance to the field”. We also thank Reviewer 4 for noting that “Highlighting the complexity and redundancy of this form of cell death is interesting.” Lastly, we thank Reviewer 5 for noting that “the conclusions that heat stress activates panoptosis in addition to apoptosis and necroptosis and that NINJ1 is involved in this process are useful to the field”.

In our revision, we have carefully considered all the reviewers’ comments and followed their expert guidance to improve the clarity of our manuscript, include additional methodological details, and add further controls to strengthen our conclusions. We have listed below new data that we have generated for the revised manuscript. These new results show that:

- Deletion of TRIF prevents the activation of pyroptosis, apoptosis, and necroptosis in response to LPS plus heat shock (HS), not HS alone.
- HS triggers assembly of a complex containing NLRP3, ASC, caspase-8, and RIPK3 to induce inflammatory cell death, and this complex can be visualized by microscopy and immunoprecipitation.
- The release of DAMPs and cytokines was reduced in cells with a combined deletion of caspase-8, RIPK3, caspase-1, and caspase-11.

We would also like to highlight that our previous revision has already demonstrated that:

- HS induces cell death which is potentiated during infection or in response to bacterial components such as LPS.
- TLR signaling specifically through TRIF, and not MyD88, enhances cell death triggered by HS during infection.
- HS triggers assembly of a complex containing NLRP3, caspase-8, and RIPK3 to induce inflammatory cell death.
- LPS plus HS induces NLRP3 inflammasome activation, but NLRP3 is not essential for the cell death.
- A screen of innate immune sensors, including ZBP1, NLRP3, AIM2, NLRP1b, NLRC4, Pyrin, NLRC1, NLRC2, NLRC3, NLRC5, NLRP6, and NLRP12, showed none of these sensors are required for LPS plus HS-mediated cell death.
- The cell death induced by LPS plus HS is characterized by activation of caspase-1 and -11, gasdermin D and E, caspase-8, -3, and -7, and MLKL, markers of inflammatory cell death.
- Deletion of molecules from pyroptosis, apoptosis, or necroptosis alone does not prevent the LPS plus HS-mediated cell death.
- *Casp1^{-/-}Casp8^{-/-}Ripk3^{-/-}* cells are partially protected, and *Casp1^{-/-}Casp11^{-/-}Casp8^{-/-}Ripk3^{-/-}* cells are completely protected from cell death induced by LPS plus HS.
- RIPK3 and caspase-7 significantly contribute to the inflammatory cell death induced by LPS plus HS.
- ZBP1 and MLKL are not involved in the inflammatory cell death induced by HS in the presence or absence of LPS.
- Among all 20,176 genes assessed in the CRISPR screen, NINJ1 is ranked 5,033 based on fold enrichment, placing it in the top 25% of all genes; furthermore, when significance is factored, NINJ1 is ranked 324, placing it in the top 2%.
- Whole genome CRISPR/CAS9 screen analysis identifies NINJ1 as a cell death executioner that drive inflammation during HS.
- NINJ1 is a critical executioner of PANoptosis in response to LPS plus HS.

- All isoforms of gasdermin C (GSDMC) are not required executioners for LPS plus HS-mediated cell death.
- GSDMC does not act redundantly with GSDMD, GSDME, or MLKL to execute cell death in response to LPS plus HS.
- LPS plus HS induces the expression and oligomerization of NINJ1.
- NINJ1 oligomerization is regulated by caspase-8 and RIPK3.
- The physiological trigger of HS combined with bacterial PAMPs induces lethality in a murine model.
- Lethality is rescued by deletion of key cell death molecules, including executioner NINJ1 or combined deletion of caspase-8, RIPK3, caspase-1, and caspase-11.

Overall, this study is the culmination of a decades-old question about innate immune mechanisms of heat stress-mediated pathology and combines an analysis of 40 genetically distinct models and a high throughput CRISPR screen to define the regulation of the innate immune response to heat stress. We have included a point-by-point response to the specific comments in the following pages. We look forward to sharing these exciting data with the field.

Reviewer #1:

I reviewed this manuscript for [redacted] earlier and disagreed with other reviewers on their pointed out weaknesses and perceived lack of novelty. This manuscript went through a high quality revision at [redacted], where the authors provided a large body of new experiments to further support their conclusion.

In my opinion, this is a very compelling and high-quality manuscript about the mechanism of heat stress induced cell death. The authors use genetic deletion or siRNA silencing of all possible key players involved in pyroptosis, apoptosis and necroptosis to dissect their respective contributions to- and to dissect the pathway of heat stress induced cell death revealing PANoptosis as the involved mechanism.

I have no additional questions and consider this manuscript to be of high quality, high importance to the field and an essential complementation and extension of earlier studies.

We greatly appreciate the time and expertise the reviewer has contributed to reviewing our study. We appreciate that the reviewer found our work to be “of high quality, high importance to the field and an essential complementation and extension of earlier studies.”

Reviewer #4:

The authors demonstrate a cell death response to heat stress alongside TLR signaling through TRIF, inducing cell death through Caspase8-Ripk3, Caspase-1/11, and Ninjurin1. However, these findings seem to conflict with previous published findings of heat stress induced cell death. No resolution or serious discussion is given on why that may be. Yet, I am missing mechanistic depth on how this pathway is triggered. Instead, the authors focus on searching for executioners of this cell death. The complex genetic analysis using a variety of mouse KO is appreciated, and the progression from individual regulators to triple KO and quadruple KO, providing more and more complete cell death reduction. Highlighting the complexity and redundancy of this form of cell death is interesting, but this cannot cover the shortcomings in explanation of how the signals are triggered. In addition, the authors make some conclusions from their cell death-Crispr screen that I don't really find well supported. Overall, this publication has substantial shortcomings. It is missing some essential controls, some necessary additional analyses, and shows a lack of depth on search for mechanism. An earlier publication by Yuan et al Science 2022 also discusses heat stress and this takes some novelty away, in addition to providing different results about signaling molecules – this is also difficult to comprehend.

We greatly appreciate the time and expertise the reviewer has contributed to reviewing our study. We appreciate that the reviewer found that the “complex genetic analysis using a variety of mouse KO is appreciated” and for noting that “[h]ighlighting the complexity and redundancy of this form of cell death is interesting”. We have carefully considered the reviewer's comments and followed the specific guidance from the editor in addressing these points to discuss the critical differences between our study and that of Yuan et al., provide additional information and interpretation of the CRISPR screen, and add the requested controls. We have described these revisions in our point by point response below.

Specific comments:

It is unsatisfying that little effort has been made to identify which mechanisms may trigger the heat related stress signaling.

We respectfully disagree with the reviewer that there was little effort to understand the mechanism triggering the heat stress (HS)-mediated signaling and cell death. In fact, we invested significant effort on this point and combined an analysis of 40 genetically distinct models and a high throughput CRISPR screen to define the regulation of the innate immune response to heat stress. Our efforts have already identified that:

- HS induces cell death which is potentiated during infection or in response to bacterial components such as LPS.
- TLR signaling specifically through TRIF, and not MyD88, enhances cell death triggered by HS during infection.
- Deletion of TRIF prevents the activation of pyroptosis, apoptosis, and necroptosis by LPS plus HS, not HS alone.
- HS triggers assembly of a complex containing NLRP3, ASC, caspase-8, and RIPK3 to induce inflammatory cell death that can be visualized by microscopy or immunoprecipitation.
- LPS plus HS induces NLRP3 inflammasome activation, but NLRP3 is not essential for the cell death.
- A screen of innate immune sensors, including ZBP1, NLRP3, AIM2, NLRP1b, NLRC4, Pyrin, NLRC1, NLRC2, NLRC3, NLRC5, NLRP6, and NLRP12, showed none of these sensors are required for LPS plus HS-mediated cell death.

- The cell death induced by LPS plus HS is characterized by activation of caspase-1 and -11, gasdermin D and E, caspase-8, -3, and -7, and MLKL, markers of inflammatory cell death.
- Deletion of molecules from pyroptosis, apoptosis, or necroptosis alone does not prevent the LPS plus HS-mediated cell death.
- *Casp1^{-/-}Casp8^{-/-}Ripk3^{-/-}* cells are partially protected, and *Casp1^{-/-}Casp11^{-/-}Casp8^{-/-}Ripk3^{-/-}* cells are completely protected from cell death induced by LPS plus HS.
- RIPK3 and caspase-7 significantly contribute to the inflammatory cell death induced by LPS plus HS.
- ZBP1 and MLKL are not involved in the inflammatory cell death induced by HS in the presence or absence of LPS.
- Among all 20,176 genes assessed in the CRISPR screen, NINJ1 is ranked 5,033 based on fold enrichment, placing it in the top 25% of all genes; furthermore, when significance is factored, NINJ1 is ranked 324, placing it in the top 2%.
- Whole genome CRISPR/CAS9 screen analysis identifies NINJ1 as a cell death executioner that drive inflammation during HS.
- NINJ1 is a critical executioner of PANoptosis in response to LPS plus HS.
- All isoforms of gasdermin C (GSDMC) are not required executioners for LPS plus HS-mediated cell death.
- GSDMC does not act redundantly with GSDMD, GSDME, or MLKL to execute cell death in response to LPS plus HS.
- LPS plus HS induces the expression and oligomerization of NINJ1.
- NINJ1 oligomerization is regulated by caspase-8 and RIPK3.
- The physiological trigger of HS combined with bacterial PAMPs induces lethality in a murine model.
- Lethality is rescued by deletion of key cell death molecules, including executioner NINJ1 or combined deletion of caspase-8, RIPK3, caspase-1, and caspase-11.
- The release of DAMPs and cytokines was reduced in cells with a combined deletion of caspase-8, RIPK3, caspase-1, and caspase-11.

I cannot consolidate Yuan, F. et al.'s finding that ZBP1 is key to HS cell death, which did not involve TRIF. With the findings here that are dependent on TRIF but do not involve ZBP1. Especially in Extended figure 4, when performing the 1hr heat stress (HS) without LPS, which appears identical to Yuan, F. et al 's HS method. Could the authors highlight differences between your methods that may explain this?

Here the reviewer is asking about methodological differences between our study and that of Yuan et al., where they found ZBP1 to be a key mediator of HS-induced cell death that did not rely on TRIF. We would like to first point out that in our study, we identified a role for TRIF only when LPS was added, not under the condition used by Yuan et al. where HS alone is the trigger. Therefore, our findings match those of the previous study for HS-mediated cell death being TRIF-independent (shown below; added to revised manuscript in Extended Data Figure 2E). Because LPS specifically engages both MyD88 and TRIF signaling, we expected a role for one of these molecules in this context. We found that cell death induced by LPS during HS is mediated by TRIF signaling, but not MyD88 signaling.

[FIGURE REDACTED]

Regarding the role for ZBP1, we initially tested the LPS plus HS condition, which differs from the HS alone trigger used in the Yuan et al. study, and found no role for ZBP1. As a follow up, we then replicated the methods used by Yuan et al., using HS alone as the cell death trigger. However, we still could not replicate the findings of Yuan et al. to see a role for ZBP1 in this cell death. Additionally, we performed ZBP1 silencing experiments and generated new *Zbp1*^{-/-} mice. Consistently, our findings using three independent lines of evidence contradict those from Yuan et al. and demonstrate that ZBP1 is not essential for cell death induced by HS. We have discussed these contrasting phenotypes more in depth in the revised manuscript.

Figure 2A: HS of 30min alone failed to activate GSDMD, caspase-1/11, or GSDME, but earlier was shown that heat stress of 30min did not cause robust cell death. So this is not surprising, it should serve as a negative control. Are the authors trying to claim that HS death without LPS is not pyroptotically driven? For that claim it would be more telling if HS that does cause robust death was used, like HS induced on a longer period of time. Then it could be analyzed for what executor molecules are responsible for cell death without LPS.

Here the reviewer is asking for clarification regarding the role of pyroptosis in the HS-mediated cell death. HS without LPS for 30 min does induce cleavage of caspase-1, caspase-11, GSDMD, and GSDME, albeit at lower levels than LPS plus HS, due to the known role of LPS in activating robust inflammasome activation. Therefore, we are not claiming that HS alone-induced cell death does not involve the activation of pyroptotic molecules. To address the reviewer's second question regarding the executioners involved in HS-mediated cell death without LPS, we tested the deletion of GSDMD, GSDME, and MLKL individually and in combination, but none of these rescued the cell death (shown below for reviewer's reference), suggesting additional executioners are involved.

[FIGURE REDACTED]

Ext. Fig. 5 G-H: The authors claim this LPS/HS condition forms one complex with NLRP3, RIPK3, and Casp8. However, NLRP3 is not probed in the co-immunoprecipitation blots and only one image of one NLRP3 speck colocalizing with Ripk3 and Caspase-8 was provided. Furthermore, in the one image provided, there are other NLRP3 specks present that do not colocalize with Ripk3 and Casp8. I'm not certain this has to be one big speck complex, as different complexes could be formed in one cell at the same time.

Here the reviewer is asking for additional evidence that NLRP3 is in the complex with RIPK3 and caspase-8 in response to LPS plus HS. To complement the colocalization of NLRP3, caspase-8, and RIPK3 we have already shown by microscopy in the previous version of our manuscript in response to LPS plus HS, we conducted co-immunoprecipitation to investigate the physical interaction of NLRP3 with caspase-8 and RIPK3. Our results demonstrate that NLRP3 and caspase-8 coimmunoprecipitated with RIPK3 in response to LPS plus HS (shown below; added to revised manuscript in Extended Data Figure 6G).

[FIGURE REDACTED]

Additionally, we agree with the reviewer that not all NLRP3 specks colocalize with RIPK3 and caspase-8, suggesting that different complexes could be formed in one cell at the same time. This is consistent with the expected cellular physiology.

Fig 3 D & F: Figure 2 blots show that HS+LPS induce GsdmD, GsdmE, and MLKL activation, but then in these figures, none of them contribute to PI uptake at all. When these molecules are activated, they make pores on the membrane, making the membrane permeable to dyes such as PI. Thus, the authors conclude that another pore must be made that can supersede all of these. However, an alternative explanation could be that these activated molecules are unable to aggregate on the cell membrane and form pores in HS conditions. Has gasdermin/MLKL oligomerization and pore formation been analyzed? This could be examined.

While we agree with the reviewer that activated pore-forming molecules could be unable to aggregate on the cell membrane and form pores in response to LPS plus HS, the presence of the membrane impermeable PI inside the cell under these conditions still suggests another pore must be forming. Deleting GSDMD, GSDME, and MLKL individually or in combination was not sufficient to inhibit cell death (shown below; Figure 3D, E in the previous submission), indicating the involvement of another molecule beyond these classical pore-forming molecules.

[FIGURE REDACTED]

In our quest to identify the critical executioner responsible for LPS plus HS-induced cell death, we have demonstrated that targeting NINJ1 through deletion or blocking its activation with glycine can effectively regulate this process (shown below; Figure 4E, F in the previous submission). Therefore, we have concluded that NINJ1 has a critical role in this cell death, while GSDMD, GSDME, and MLKL do not. The oligomerization status of

these molecules does not affect this conclusion and is beyond the scope of this manuscript.

[FIGURE REDACTED]

Fig 4A: Crispr cell death screen: Much of these data are difficult to understand, including the evaluation put forward in the paper. First, I am trying to find out how the screen was done, and the various descriptions appear incomplete. Information about this needs to be clearly stated. Importantly - what is the read-out in the screen? If it is PI, then Ninj1 should not show up as it regulates larger cell ruptures and not PI permeability. Ninj1 is not a topmost enriched gene as it's ranked 5000+ in fold-change and 300+ by p-value. GsdmC isoforms are also ranked 300+ by p-value and in the thousands by fold- change. So, it doesn't appear that this whole genome CRISPR screen supports the claims made. It is then not surprising that silencing all isoforms of GsdmC failed to provide protection. I would say that the interpretations from the authors on this aspect are not supported well. I don't think it is possible to identify Ninj1 from the screen alone.

As suggested by the reviewer, we have added additional details on how the CRISPR screen was performed and interpreted to identify regulatory genes in the Results and Methods sections of the revised manuscript:

To identify other cell death executioners and regulators involved in response to LPS plus HS, we next performed a whole genome CRISPR-Cas9 knockout screen in murine immortalized BMDMs (iBMDMs), a well-established and widely used cell type for studying innate immune mechanisms of cell death and inflammation. Using iBMDMs offers advantages over using primary BMDMs because iBMDMs can be produced in a nearly unlimited supply of cells, which is critical for large scale analyses. Cas9-expressing iBMDMs were generated from Cas9-GFP knock-in mice. A total of 300×10^6 Cas9-iBMDMs were distributed across twelve 15 cm^2 tissue culture dishes at 25×10^6 cells per dish and infected with the Brie library of lentiviral particles, which carry four gRNAs for each gene to cover the entire genome, at an MOI of 0.3 in 25 ml of complete DMEM (DMEM supplemented with 10% heat-inactivated fetal bovine serum (HI-FBS; S1620, Biowest)) with 100 ml of LentiBOOST transduction reagent (SB-P-LV-101-02, Lentivirus Transduction Enhancer Solution, Siron Biotech). After infection, the iBMDMs were incubated for 24 h for efficient transduction. These transduced cells were expanded with intermittent passaging to avoid overcrowding of the cells and to generate a sufficient number of cells for the downstream whole-genome CRISPR screens. Two replicates of an adequate number of cells were used as control to obtain a representation (screen depth) of > 500 cells for each sgRNA of the library, and a similar

number of cells from the same batch of virus preparation were stimulated with LPS priming for 2 h, then incubated at 43 °C for 1 h to induce HS. After 24 h, the media was removed and cells in the unstimulated and stimulated conditions were washed with PBS to remove the non-adherent dead cells, leaving only the adherent surviving cells for downstream analyses. The surviving cells from the unstimulated and LPS plus HS-treated samples were then subjected to CRISPR screen enrichment analysis. Total genomic DNA was isolated using NucleoSpin® Blood kits (Takara Bio Inc., USA; 740954 and 740950) and the concentrations of the isolated gDNA samples were measured using NanoDrop (Thermo Fisher Scientific, USA).

Next-generation sequencing (NGS) of the PCR-amplified, barcoded gRNAs was performed to quantitatively identify gRNAs that were enriched in the surviving pool of cells. Because the presence of a gRNA should delete the corresponding gene in that cell, enriched gRNAs were expected to represent genes where deletion rescues the cells from LPS plus HS-mediated cell death, suggesting they are positive regulators of the cell death.

Using the MAGeCK pipeline, the log₂fold change was estimated with significance levels for the genes in the CRISPR screen. The genes with positive fold change were expected to be important for cell death. The top gene hits along with their significance from the CRISPR screen were highlighted in a volcano plot using MAGeCKFlute v2.0.0 (ref. ⁸⁴).

Fig 4E-J: As for *Ninj1*, *Ninj1* is necessary for LDH and DAMP release, as well as lysis of the cell. But in Kayagaki et al. in Nature 2021 where this was discovered, *Ninj1* was not responsible for membrane permeability in uptake of YOYO-1, or the release of IL-1b. Showing equivalent YOYO-1 uptake as WT cells when undergoing pyroptosis by LPS electroporation. Thus, it is surprising that in LPS+HS and HS conditions, *Ninj1* deletion resulted in a significant reduction in PI uptake. I cannot understand this discrepancy and it should be carefully discussed. It would be highly informative then to have a negative control in these PI uptake experiments, and at the very least a no treatment condition. At minimum

Here the reviewer is requesting a negative control for the PI uptake experiments to confirm that NINJ1 is responsible for the pore formation to allow PI to enter. As suggested, we have now included a condition with no treatment (PI added without HS or LPS plus HS) in the NINJ1 analysis to compare PI uptake by cells under basal conditions (shown below: Figure 4E, G in the revised manuscript). These results show that PI was only entering the cells in response to HS or LPS plus HS, not the media control, and that deletion of *Ninj1* significantly reduced the cell death.

[FIGURE REDACTED]

Additionally, our paper is not the first to show a significant reduction in PI uptake upon NINJ1 deletion. Igor Brodsky also showed decreased PI uptake upon NINJ1 deletion during *Yersinia* infection and Staurosporine treatment (PMID: 34648590). Therefore, it seems that differences in the underlying stressors triggering cell death may account for some of the observed variations. There is also a possibility that NINJ1 could influence the oligomerization or regulatory functions of other pore-forming executioners. We have included a new discussion addressing this potential aspect.

Furthermore, it appears that Ninj1 deletion did not fully suppress cell death, though it is hard to tell without comparing to no treatment. This may imply that Ninj1 is not solely responsible for executing cell death by HS, but plays a large significant role in its membrane permeability.

We agree with the reviewer that NINJ1 deletion did not fully suppress cell death, indicating that NINJ1 is not solely responsible for executing cell death in response to HS. We also noted reduced cell death in *Casp7*^{-/-} cells (Figure 3A-3C), indicating that caspase-7 partially contributes to cell death through a mechanism distinct from NINJ1. We have discussed this in the revised manuscript.

Fig 5D: Why is there no ladder provided?

As requested by the reviewer, we have added information about the protein size in Figure 5D (shown below) and indicated the use of the NativeMark unstained protein ladder in the Methods section of the revised manuscript.

[FIGURE REDACTED]

Fig 6: The authors portray these experiments as modeling a patient who is fighting off infection with an uncontrollable fever. However, mice lower their body temperature when they are undergoing inflammation and infection. Thus, forcing their body temperature over base body temperature while undergoing LPS shock may have vast effects on their immune system and thermoregulation across bodily systems. They inject IP LPS to induce shock (5mg/kg LPS) and place them under heat stress of 39C for 2hrs. An analysis of what this does to the body and the

immune system, such as by analyzing electrolytes, cytokines, and the activity of immune cells would be informative and necessary to understand the physiology and the context. Comparing these metrics to heat stress without LPS would provide better information about the setting. Though in vivo protection does partially line up with their proposed pathway of Caspase-8-Ripk3 and Ninj1, without additional data it is hard to define if these results are equivalent to those demonstrated by BMDMs in vitro. At least the T-KO and Q-KO giving 100% survival does indicate that survival is regulated by caspases. But additional data is required to complete our understanding of what is occurring in this model.

As suggested by the reviewer, we have included additional analyses from our in vivo heat shock model. We observed a reduction in the release of HMGB1, LDH, IL-1 β , and IL-18 in the serum of *Casp1*^{-/-}*Casp8*^{-/-}*Ripk3*^{-/-} (TKO) and *Casp1*^{-/-}*Casp11*^{-/-}*Casp8*^{-/-}*Ripk3*^{-/-} (QKO) mice, indicating that the mortality rate induced by LPS plus heat stress correlated with the excessive release of DAMPs and cytokines (shown below; Figure 6C–6F in the revised manuscript). We have included these new data figures and provided additional results, legends, and methods in the appropriate sections of the revised manuscript.

[FIGURE REDACTED]

The discussion proposes cytokine storms as a cause for the pathological effects of HS. However, the paper does little to no investigation of cytokine release and regulation. Only in Extended Fig. 7 are cytokines even measured, and only IL-1b and IL-18 in one in vitro experiment. More should be done on measuring cytokines throughout, such as by ELISA.

As suggested by the reviewer, we have included additional analyses from our in vivo heat shock model. We observed a reduction in the release of HMGB1, LDH, IL-1 β , and IL-18 in the serum of *Casp1*^{-/-}*Casp8*^{-/-}*Ripk3*^{-/-} (TKO) and *Casp1*^{-/-}*Casp11*^{-/-}*Casp8*^{-/-}*Ripk3*^{-/-} (QKO) mice, indicating that the mortality rate induced by LPS plus heat stress correlated with the excessive release of DAMPs and cytokines (shown above; Figure 6C–6F in the revised manuscript). We have included these new data figures and provided additional results, legends, and methods in the appropriate sections of the revised manuscript.

Extended figure 2 demonstrates that TLR3/TLR4 responses are the most robust in causing cell death under heat stress. Usually Poly(I:C) and LPS provide priming for secondary signals, but provide little cell death on their own, yet these primed, heat stressed cells are directly going into programmed death. Heat stress can cause a wide range of effects on the cell. So it is curious why TLR3 and TLR4 respond so robustly to this heat stress induced death, but not the other TLRs tested.

Here the reviewer is asking for clarification as to why TLR3 and TLR4 are the most important to induce the HS-mediated cell death. Among TLR agonists, only poly(I:C) (TLR3) or LPS (TLR4) were observed to increase cell death during HS. Given that both TLR3 and TLR4 mediate signaling through TRIF, we hypothesized that TRIF played a more important role in the cell death than MyD88. We tested this and found that the cell death induced by LPS plus HS was TRIF-dependent (shown below; Extended Figure 2C, D in the revised manuscript). The critical role of TRIF likely explains why TLR3/TLR4 signaling is more important to prime the cell death.

[FIGURE REDACTED]

TRIF signaling of course is the easiest explanation, and the KOs confirm the expectation that TRIF is important for LPS+HS. Yet TRIF is early in TLR3/TLR4 signaling cascade, and it is hard to dissociate what molecules are affected by HS, and if it is signaling through the expected TRIF pathway. Could the authors address this?

Here the reviewer is asking for further clarification regarding the role of TRIF in the downstream signaling activated by LPS plus HS. We have now assessed the effect of TRIF deletion on the activation of the downstream cell death molecules, and we found that TRIF deletion inhibits the activation of the PANoptosis pathway (shown below; Extended Data Figure 5A–C in the revised manuscript), suggesting that TRIF is a key upstream regulator of the full signaling pathways. We have included these data in the results and legends in the appropriate sections.

[FIGURE REDACTED]

Minor comments:

Perhaps testing of other TRAF related responses independent of TLRs, such as TNFR1, ILR1, or RigI, would be informative. As well as looking at TRAF3, other TRAFs, TBK1, or IRF3.

We greatly appreciate the reviewer's valuable comment. However, we believe that determining the role of TRAFs during heat stress is beyond the scope of this study.

Later the authors mention 1hr of HS generates similar cell death in BMDMs to that seen in LPS primed cells. Why not test if this cell death is also TRIF dependent?

While TRIF played an important role in the cell death induced by LPS plus HS, we found that TRIF deletion did not reduce cell death solely caused by HS (shown below; Extended Data Figure 2C–F in the revised manuscript). This indicates that infection or LPS in conjunction with heat stress induces distinct mechanisms compared to HS alone.

LPS plus HS-mediated cell death

[FIGURE REDACTED]

HS-mediated cell death

[FIGURE REDACTED]

Notably, the activation of caspase-8, the main regulator of apoptotic cell death induced by HS, was not significantly affected by TRIF deletion in response to HS alone, while caspase-8 activation was drastically reduced in the TRIF knockout in response to LPS plus HS (shown below; Extended Data Figure 5B, D in the revised manuscript). These results suggest that the mechanisms of cell death induced by HS, with or without infection, differ. We have included these data in the results and legends in the appropriate sections.

[FIGURE REDACTED]

Reviewer #5:

I was asked to review whether Rev #3's questions have been addressed. Upon investigating questions raised by both Rev #3 and #2, and the answers by the authors, I would like to present my analysis below.

The reviewers' questions were largely about the novelty of the study and whether the study provided sufficient advance over previously published papers to warrant its publication in **[redacted]**. I imagine that the authors then transferred their study to Nat Comm by addressing the comments. In my opinion, the study did have some overlap with previous papers, but because NINJ1 is still quite a new molecule implicated in membrane rupture post cell death, the new insights would still serve the field well. The publication in a lower tier, but still very good, journal of this study would further expand the interest of the field in this molecule. In particular, the conclusions that heat stress activates panoptosis in addition to apoptosis and necroptosis and that NINJ1 is involved in this process are useful to the field. I also think that the in vivo data on NINJ1 KO adds additional insights to the current understanding. Thus, I support the publication of the data.

We greatly appreciate the time and expertise the reviewer has contributed to reviewing our study. We appreciate that the reviewer found that “the new insights would still serve the field well”, that “the in vivo data on NINJ1 KO adds additional insights to the current understanding”, and that the reviewer “support[s] the publication of the data.”

REVIEWERS' COMMENTS

Reviewer #4 (Remarks to the Author):

The manuscript is improved upon revision and I appreciate that the authors have added new data and text. A few remaining points:

1- Trigger of heat related stress signaling.

All of these comments in the rebuttal are about how LPS triggers Trif, but not about the mechanism of how heat stress itself initiates signaling. Adding information on this would help the publication.

2 - Fig 4A Crispr cell death screen.

The methods section for this is a necessary addition. Still it is unsatisfactory to claim Ninj1 and GsdmC are hits from this screen, however. These “hits” could have easily been picked out of the literature without the screen, which only partially supports them, being far down on the list.

3 - Fig 6 Mouse model evaluation.

The paper would benefit from more data evaluating the model overall, to my knowledge this is a new mouse model, and therefore should be given a proper evaluation. Cytokine data gives more information about the immune response but doesn't necessarily evaluate how the heat stress is affecting the mouse physiology.

4- Ext Fig 2

There is still a lack of clarity on what is the secondary signal induced by heat in this cell death pathway. This is linked back to comment 1.

We greatly appreciate the time and expertise the reviewers have contributed to the peer review process. We are grateful for the comments from Reviewer 4 noting that “The manuscript is improved upon revision and I appreciate that the authors have added new data and text.” In our revision, we have carefully considered the remaining comments from Reviewer 4 and followed the expert guidance of the Reviewer to improve the clarity of our manuscript.

We are excited to share this study with the broader scientific community. Overall, this study is the culmination of a decades-old question about innate immune mechanisms of heat stress-mediated pathology and combines an analysis of 40 genetically distinct models to define the regulation of the innate immune response to heat stress. We have included a point-by-point response to the specific comments in the following pages.

Reviewer #4:

The manuscript is improved upon revision and I appreciate that the authors have added new data and text.

We greatly appreciate the time and expertise the reviewer has contributed to reviewing our study.

A few remaining points:

1- Trigger of heat related stress signaling. All of these comments in the rebuttal are about how LPS triggers Trif, but not about the mechanism of how heat stress itself initiates signaling. Adding information on this would help the publication.

text redacted

FIGURE REDACTED

2 - Fig 4A Crispr cell death screen. The methods section for this is a necessary addition. Still it is unsatisfactory to claim Ninj1 and GsdmC are hits from this screen, however. These “hits” could have easily been picked out of the literature without the screen, which only partially supports them, being far down on the list.

We respectfully disagree with the reviewer regarding whether *Ninj1* and *GsdmC* are “hits” in our CRISPR screen. The identification of “hits” in CRISPR screens is based on statistical analysis, and we have followed the standard approach to analyze our results and select the hits. In this study, we used the CRISPR screen as a starting point to identify a large number of molecules that may have a role in the cell death pathway occurring in response to heat stress and PAMPs. Based on fold enrichment analysis, *NINJ1* is ranked 5,033, placing it in the top 25% of all genes. However, consideration of the *P* value is critical to filter out false positive results, and adding this parameter ranks *NINJ1* as 324, placing it in the top 2% of all genes. Furthermore, given that our analysis was focused on trying to identify the molecules involved in driving the cell death in response to heat stress, we filtered the results to focus on the molecules involved in the execution of inflammatory cell death, as described in our manuscript. Using this approach, we identified *NINJ1* and *GSDMC* as the top CRISPR hits, with *NINJ1* being the number 1 hit (shown below).

[FIGURE REDACTED]

We then performed *in vitro* analyses to validate the cell death executioners identified in the CRISPR screen. We found that *GSDMC* had no role in the cell death (shown below).

[FIGURE REDACTED]

In contrast, we found a clear role for *NINJ1* in cell death induced by heat stress, as shown by a reduction in cell death in cells from a newly generated *Ninj1* KO mouse and reduced cell death in response to the *NINJ1* inhibitor glycine (shown below).

[FIGURE REDACTED]

Based on these in vitro results, we also performed an in vivo assessment to conclusively test whether NINJ1 plays a role in the pathophysiology of heat stress. We found that we found that NINJ1-mediated cell death significantly contributes to heat stress-induced mortality (shown below).

[FIGURE REDACTED]

Overall, these findings show a clear role for NINJ1 in the inflammatory cell death and pathophysiology induced by heat stress plus PAMPs. Therefore, our CRISPR analysis laid a solid foundation for these subsequent validations to provide multiple lines of evidence supporting the role of NINJ1, making this analysis an essential component of our study. Furthermore, this CRISPR screen is a powerful tool to identify molecules at every stage of the cell death process, including upstream regulators, effectors, and downstream executioners. We feel that the inclusion of these data also allows other investigators to identify their molecules of interest and probe further into their functions in follow up studies.

In the revised manuscript, we have not referred to these molecules as “hits” and instead refer to them as “the topmost enriched genes based on P value” among the cell death executioners for clarity.

3 - Fig 6 Mouse model evaluation. The paper would benefit from more data evaluating the model overall, to my knowledge this is a new mouse model, and therefore should be given a proper evaluation. Cytokine data gives more information about the immune response but doesn't necessarily evaluate how the heat stress is affecting the mouse physiology.

As suggested by the reviewer, to confirm that the heat stress was inducing a physiological response, we also performed additional analyses from our in vivo heat shock model. We observed a significant increase in serum levels of alanine aminotransferase (ALT) and aspartate aminotransferase (AST) in mice subjected to LPS plus heat stress, which is indicative of liver injury. The levels of ALT and AST were reduced in the serum of *Casp1^{-/-}Casp8^{-/-}Ripk3^{-/-}* (TKO) and *Casp1^{-/-}Casp11^{-/-}Casp8^{-/-}Ripk3^{-/-}* (QKO) mice, indicating that the liver injury rate induced by LPS plus heat stress was decreased in these animals (shown below). However, due to space limitations, these data have not been included in the revised manuscript.

[FIGURE REDACTED]

4- Ext Fig 2. There is still a lack of clarity on what is the secondary signal induced by heat in this cell death pathway. This is linked back to comment 1.

text redacted

FIGURE REDACTED